# Retrotransposon instability dominates the acquired mutation landscape of mouse induced pluripotent stem cells

Patricia Gerdes [1,10], Sue Mei Lim[2,3,4,10], Adam D. Ewing [1,10], Michael R. Larcombe[2,3,4], Dorothy Chan [1], Francisco J. Sanchez-Luque[1,5], Lucinda Walker[1], Alexander L. Carleton[1], Cini James[1], Anja S. Knaupp[2,3,4], Patricia E. Carreira [1], Christian M. Nefzger [2,3,4], Ryan Lister [6,7], Sandra R. Richardson [1] ✉, Jose M. Polo [2,3,4,8] ✉ & Geoffrey J. Faulkner [1,9] ✉

Induced pluripotent stem cells (iPSCs) can in principle differentiate into any cell of the body, and have revolutionized biomedical research and regenerative medicine. Unlike their human counterparts, mouse iPSCs (miPSCs) are reported to silence transposable elements and prevent transposable element-mediated mutagenesis. Here we apply short-read or Oxford Nanopore Technologies long-read genome sequencing to 38 bulk miPSC lines reprogrammed from 10 parental cell types, and 18 single-cell miPSC clones. While single nucleotide variants and structural variants restricted to miPSCs are rare, we find 83 de novo transposable element insertions, including examples intronic to *Brca1* and *Dmd*. LINE-1 retrotransposons are profoundly hypomethylated in miPSCs, beyond other transposable elements and the genome overall, and harbor alternative protein-coding gene promoters. We show that treatment with the LINE-1 inhibitor lamivudine does not hinder reprogramming and efficiently blocks endogenous retrotransposition, as detected by long-read genome sequencing. These experiments reveal the complete spectrum and potential significance of mutations acquired by miPSCs.

Induced pluripotent stem cells (iPSCs) resemble embryonic stem cells (ESCs) in their near unlimited capacity for self-renewal and differentiation potential[1]. These properties have driven widespread uptake of iPSCs in clinical and research applications[2–4]. Despite their immense therapeutic promise, the reprogramming process required to generate iPSCs can produce genomic and epigenomic aberrations[4–8]. These abnormalities could undermine the functional equivalence of iPSCs and ESCs, or alter the phenotype of iPSC-derived differentiated cells, and hence necessitate genetic and functional screening of iPSCs prior to their use in the clinic[9]. Fortunately, whole genome sequencing (WGS) analyses of single nucleotide variants (SNVs), copy number variants, and structural variants (SVs) restricted to human and mouse

[1]Mater Research Institute - University of Queensland, TRI Building, Woolloongabba, QLD 4102, Australia. [2]Department of Anatomy & Developmental Biology, Monash University, Melbourne, VIC 3800, Australia. [3]Development and Stem Cells Program, Monash Biomedicine Discovery Institute, Melbourne, VIC 3800, Australia. [4]Australian Regenerative Medicine Institute, Monash University, Melbourne, VIC 3800, Australia. [5]GENYO. Pfizer-University of Granada-Andalusian Government Centre for Genomics and Oncological Research, PTS, Granada 18016, Spain. [6]Australian Research Council Centre of Excellence in Plant Energy Biology, School of Molecular Sciences, The University of Western Australia, Perth, WA 6009, Australia. [7]Harry Perkins Institute of Medical Research, Perth, WA 6009, Australia. [8]Adelaide Centre for Epigenetics and The South Australian Immunogenomics Cancer Institute, Faculty of Health and Medical Sciences, The University of Adelaide, Adelaide, SA 5005, Australia. [9]Queensland Brain Institute, University of Queensland, Brisbane, QLD 4072, Australia. [10]These authors contributed equally: Patricia Gerdes, Sue Mei Lim, Adam D. Ewing. ✉e-mail: sandra.richardson@mater.uq.edu.au; jose.polo@monash.edu; faulknergj@gmail.com

iPSC lines have found relatively few conclusive reprogramming-associated mutations[10–12]. Instead, most mutations acquired by iPSCs appear to occur before and after reprogramming[10,11,13], implying they are not caused by molecular processes intrinsic to iPSC generation. Transposable elements (TEs) may present an important exception to this rule, where the attainment of a pluripotent state via reprogramming leaves iPSCs vulnerable to TE-mediated mutagenesis.

The retrotransposon long interspersed element 1 (LINE-1, or L1) is active in nearly all mammals[14]. L1 autonomously mobilizes via a copy-and-paste process called target-primed reverse transcription (TPRT), which involves reverse transcription of L1 mRNA in cis, and is characterized by the generation of target site duplications (TSDs) upon L1 integration[15–20]. The C57BL/6 mouse reference genome contains ~3000 potentially mobile L1 copies belonging to three subfamilies ($T_F$, $G_F$ and A) defined by their monomeric 5′ promoter sequences, in addition to several active endogenous retrovirus (ERV) and short interspersed element (SINE) families[21–23]. By contrast, only ~100 mobile L1s from the transcribed subset Ta (-Ta)[24] subfamily are present in each individual human genome, with the vast majority of retrotransposition potential concentrated in fewer than 10 of these elements[25,26]. Perhaps owing to the disparate count of mobile TEs in each species, the rate of L1 mobilization in the mouse germline is estimated to be at least an order of magnitude higher than that of humans[27–30].

TE mobility is regulated by DNA methylation and histone modifications, as well as various post-transcriptional and post-translational mechanisms[31–41]. Reprogramming somatic cells to generate human iPSCs (hiPSCs) and mouse iPSCs (miPSCs) leads to epigenome-wide remodeling, including broad de-repression of L1 promoters[7,42–47]. L1 mRNA abundance increases strongly during reprogramming, and remains approximately 10-fold higher in cultured miPSCs than in parental mouse embryonic fibroblasts (MEFs)[46]. As a corollary, the early mouse embryo is a major niche for new heritable L1 retrotransposition events[28]. Mouse ESCs cultured in standard media containing serum, or naïve ground state media incorporating two small-molecule kinase inhibitors (2i), permit genome-wide L1 hypomethylation, express endogenous L1 proteins and support engineered L1 mobilization[37,41,48,49]. Engineered and endogenous L1 retrotransposition are supported by hiPSCs and human ESCs[45,50–52]. These observations collectively suggest L1 hypomethylation may be an inherent aspect of pluripotency accentuated by the molecular roadmap to an induced pluripotent state. Consequently, miPSCs are likely to harbor de novo retrotransposition events. However, a prior WGS analysis of 3 miPSC lines, employing paired-end 42mer reads and ~11× genome-wide sequencing depth, found no de novo TE insertions, and concluded that endogenous retrotransposition did not occur during miPSC production[12]. The apparent lack of TE mobility in this context remains an unresolved and yet potentially important source of miPSC mutagenesis[4]. Here, we analyze a diverse panel of miPSC genomes with short- or long-read sequencing and, as reported for hiPSCs and other pluripotent human cells[45,50–52], we detect numerous de novo TE insertions acquired by miPSCs.

## Results

### Mutational spectra of bulk miPSC populations generated from diverse cell lineages

To survey genomic variation among miPSC lines generated from a broad range of parental cell types, we bred triple transgenic C57BL/6×129S4Sv/Jae animals carrying a GFP reporter knocked into the *Oct4* locus (*Oct4*-GFP), a transcriptional activator (m2rtTA) under the control of the ubiquitously expressed Rosa26 locus (R26-m2rtTA), and a doxycycline-inducible polycistronic reprogramming cassette (Col1a1-tetO-OKSM)[53]. From each of three animals (labeled A67, A82 and A172), we used fluorescence activated cell sorting (FACS) and a range of surface markers to isolate nine isogenic primary cell populations, including three representing each germ layer (Fig. 1a). Bulk cultures

were then treated with doxycycline to induce reprogramming, followed by FACS to purify *Oct4*-GFP⁺ miPSCs. Twenty-six miPSC lines were expanded and cultured in standard media containing serum (Supplementary Data 1 and Supplementary Fig. 1). Illumina paired-end 150mer read WGS (~41× average genome-wide depth) was then applied to each miPSC line at passage 4 (p4), as well as to 3 MEF genotypic controls (Supplementary Data 1).

Concordant SNVs detected by GATK HaplotypeCaller version 3.7 and freebayes[54,55] version 1.3.1 were filtered to remove known mouse strain germline variants[56], yielding 3,603 SNVs private to a single miPSC line (average ~140 per line) and absent from the corresponding MEF samples (Supplementary Data 2). Of these, 27 in total were nonsynonymous exonic mutations. We then called concordant SVs using Delly and GRIDSS[57,58], finding 34 private SVs (~1 per line). These included a 210kbp deletion of the de novo methyltransferase *Dnmt3a* in miPSCs derived from the hematopoietic stem cells of animal A172 (Supplementary Data 2). Considering private SNVs and SVs together, we observed no significant ($p < 0.05$, one-way ANOVA with Tukey's multiple comparison test) difference in miPSC variant counts associated with parental cell type or germ layer, and SNV and SV rates resembled those found previously for fibroblast-derived miPSCs[10,12]. Choice of primary cell type, at least among the diverse panel assembled here, may therefore not significantly impact the frequency of SNVs and SVs later found in miPSC lines.

### Bulk miPSC populations harbor de novo L1 insertions

As de novo TE insertions can be overlooked by generalized SV calling algorithms[59], we used TEBreak[60] to identify non-reference TE insertions. Known non-reference genome TE insertions[56], and those found in MEF genotypic controls or multiple miPSC lines, were filtered, leaving 4 putative de novo L1 $T_F$ insertions (Fig. 1b–d, Table 1, Supplementary Fig. 2, Supplementary Data 3). To achieve even greater coverage of potentially active TEs, we performed mouse retrotransposon capture sequencing (mRC-seq), which uses sequence capture probes to enrich Illumina libraries for the 5′ and 3′ genomic junctions of mobile TEs, including $T_F$, $G_F$ and A subfamily L1s, B1 and B2 SINEs, and IAP and ETn ERVs (Supplementary Data 1)[28,61]. The combination of WGS and mRC-seq identified an additional 4 putative de novo L1 $G_F$ and $T_F$ insertions (Supplementary Fig. 3, Table 1 and Supplementary Data 3).

We PCR amplified and fully characterized each putative L1 insertion sequence. Six events were full-length, retaining 2–7 monomers at their 5′ end, and could only be PCR amplified in the miPSC line where they were detected by genomic analysis (Fig. 1b–d, Supplementary Fig. 2 and Supplementary Fig. 3). An additional L1 (labeled miPSC_6_L1) was very heavily 5′ truncated and confirmed by PCR to be private to one miPSC line (Supplementary Fig. 3). The final example (miPSC_2_L1) was heavily 5′ truncated and inverted[62] and could be PCR amplified in 7/9 miPSC lines representing all 3 germ layers of animal A67 (Fig. 1e and Supplementary Fig. 2). miPSC_2_L1 most likely represented a mosaic insertion that arose early in the embryonic development of animal A67, as found previously[28,30,63,64]. Each insertion carried TSDs of 13–19 nt, a long and pure 3′ polyA tract, and integrated at a degenerate L1 endonuclease recognition motif (5′-TTTT/AA-3′) (Table 1). These hallmarks were consistent with bona fide TPRT-mediated L1 retrotransposition events[15,16,19,65,66]. Including the mosaic miPSC_2_L1 insertion, 10/26 miPSC lines harbored at least one PCR validated de novo L1 insertion. Not counting miPSC_2_L1, miPSCs from all 3 animals and 4/9 primary cell types, representing each germ layer, presented at least one de novo L1 insertion (Fig. 1a and Supplementary Data 3). Of these insertions, 4/7 were detected in astrocyte-derived miPSCs and 6/7 in miPSCs obtained from primary cells in the bottom 50% of reprogramming efficiencies (Fig. 1a), though neither of these proportions were statistically significant (binomial test). Notably, down-sampling to 11× depth WGS, as per[12], indicated an expected 95% probability of

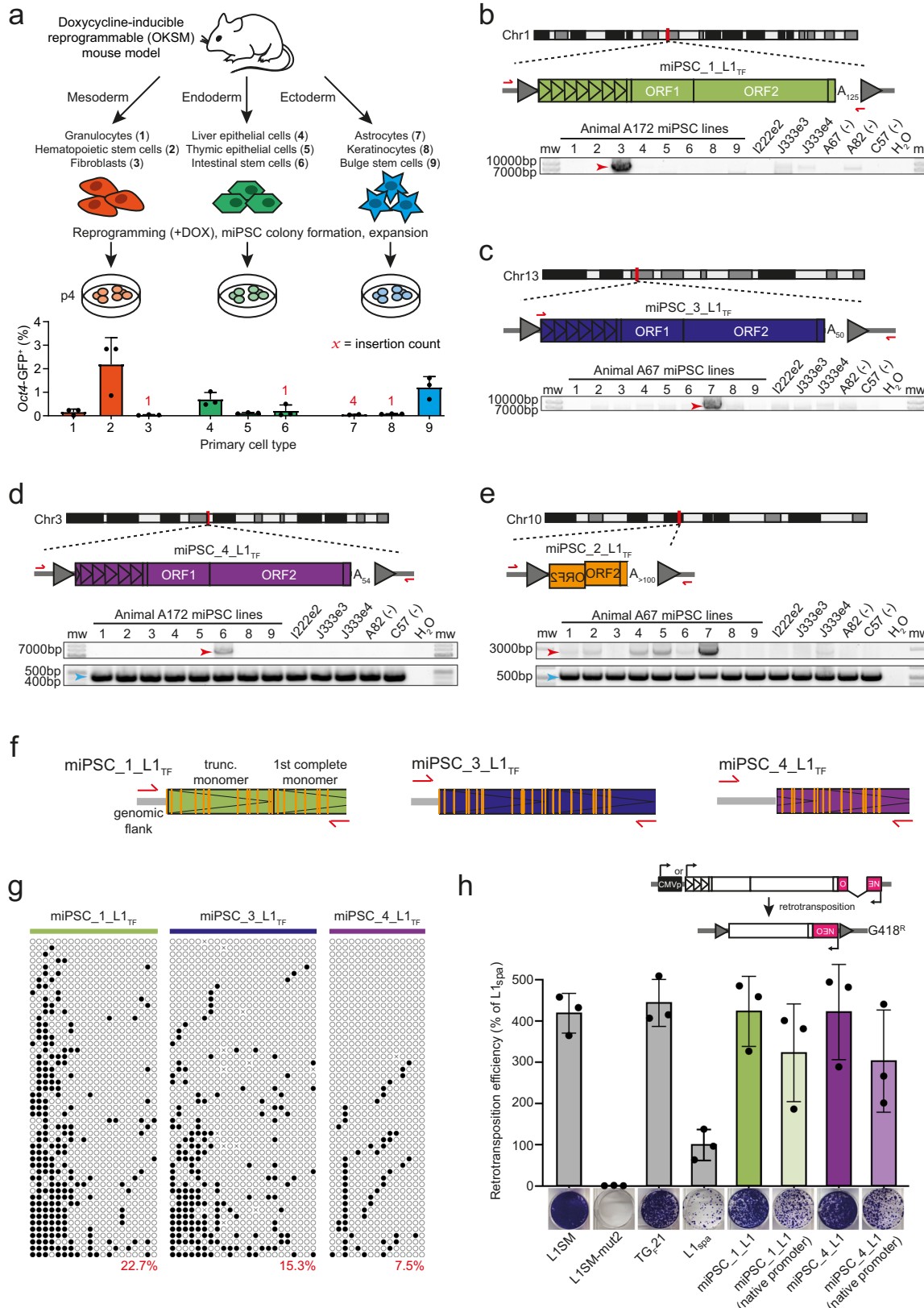

finding none of the validated de novo insertions (Supplementary Fig. 4a).

Comprehensive capillary sequencing of the 3 full-length insertions (miPSC_1_L1, miPSC_3_L1 and miPSC_4_L1) revealed that each had intact ORFs (Fig. 1b–d). To assess the potential for further mobilization of these newly retrotransposed elements, we first used multiplexed L1

locus-specific bisulfite sequencing[34,61] to measure CpG methylation of their most 5′ promoter monomers (Fig. 1f). All 3 full-length insertions were fully unmethylated in a subset of miPSCs, and their methylation decreased with distance from the L1 5′ end (Fig. 1g). Next, we cloned and tested miPSC_1_L1 and miPSC_4_L1 in a cultured cell retrotransposition assay[19,67], using the natural elements L1spa (TF

**Fig. 1 | De novo L1 insertions in germ layer specific bulk expanded miPSC lines.**
**a** Experimental design of bulk miPSC generation using a Col1a1-tetO-OSKM mouse model containing a doxycycline-inducible reprogramming cassette. Tissues were isolated and sorted by FACS to obtain 9 primary cell types (named and numbered 1–9) from each of 3 mice (A67, A82, A172). After reprogramming with doxycycline (+DOX), *Oct4*-GFP positive miPSCs were sorted and expanded in cell culture. DNA was extracted from miPSCs, sequenced via WGS and mRC-seq, and analyzed for de novo TE insertions with TEBreak. The histogram displayed underneath indicates the mean reprogramming efficiency (*Oct4*-GFP$^+$) ± SD for each primary cell type, with individual replicate values marked by black dots. Red numerals indicate the count of reprogramming-associated L1 insertions found in the miPSCs obtained from each primary cell type. Note: astrocyte-derived miPSCs were not produced for animal A172. **b** A full-length (6.8 monomers) intergenic de novo L1 $T_F$ insertion. Promoter monomers are shown as triangles within the L1 5′UTR. PolyA ($A_n$) tract length is indicated immediately 3′ of the L1. Target site duplications (TSDs) are depicted as gray arrows flanking the L1. PCR validation primers are shown as red arrows. A PCR validation agarose gel containing the full-length PCR product (red arrow) only in the fibroblast-derived miPSC line where the L1 was detected by genomic analysis is shown. miPSC line numbers are provided in (**a**). Molecular weight (mw) markers are provided at the left of the gel. Note: given the variable, and in some cases very low, reprogramming efficiencies shown in (**a**), and the objective to obtain a full set of 9 miPSC lines from each animal, we entirely used each sorted primary cell population for reprogramming, relying on PCR amplification in a single miPSC line to validate reprogramming-associated retrotransposition events. DNA from other animals included in the study are however shown at right as additional controls. **c** As for (**b**), except for an L1 $T_F$ with 5.8 promoter monomers. Note: the very faint, smaller-sized gel band observed in most of the samples was an off-target product. **d** As for panel (**b**), except for an L1 $T_F$ with 5.3 promoter monomers, and using an empty/filled PCR design where both primers are outside of the L1

insertion, generating filled L1 (red arrow) and empty wild-type (blue arrow) products. **e** As for panel (**b**), except showing a 5′ truncated and inverted/deleted L1 $T_F$ insertion and using an empty/filled PCR validation design, as per (**d**). **f** Locus-specific methylation analysis schematic representation for 3 full-length de novo L1 insertions (panels **b**–**d**). After bisulfite conversion, the 5′ monomeric sequences of each L1 were PCR amplified using primer pairs (red arrows) specific to that locus. Amplicons were then pooled and sequenced as 2×300mer Illumina reads. Orange strokes indicate CpG dinucleotides covered by the assay. **g** Methylation of the 3 L1 promoter sequences shown in panel (**f**), in the miPSC line where each de novo L1 insertion was identified. Each cartoon panel corresponds to an amplicon and displays 50 non-identical sequences (black circle, methylated CpG; white circle, unmethylated CpG; ×, mutated CpG) extracted at random from a much larger pool of available Illumina reads. The percentage of methylated CpG is indicated in the lower right corner of each cartoon in red. **h** Top: Rationale of a cultured cell retrotransposition assay[19, 67]. A mouse L1 driven by its native promoter alone, or with the addition of a CMV promoter (CMVp), is tagged with an antisense orientated neomycin (NEO) reporter cassette interrupted by an intron. Cells harboring this construct become neomycin (G418) resistant upon retrotransposition. bottom: Retrotransposition assays conducted in HeLa cells. Constructs included: L1SM[68], a highly mobile synthetic L1 (positive control); L1SMmut2, L1SM with endonuclease and reverse transcriptase active site mutations (negative control); TG$_F$21, a mobile L1 $G_F$ element;[21] L1$_{spa}$, a mobile L1 $T_F$ element;[22] miPSC_1_L1 (panel **b**); miPSC_4_L1 (**d**). Data were normalized to L1$_{spa}$ and are shown as mean ± SD of three independent biological replicates (black dots, $n = 3$), each of which comprised three technical replicates. Representative well pictures are shown below each construct. Note: L1SM retrotransposed very efficiently, leading to cell colony crowding in wells, and a likely underestimate of retrotransposition. Unless otherwise stated, L1 constructs incorporated a CMVp element. Source data are provided as a Source Data file.

---

subfamily)[22] and TG$_F$21 (G$_F$ subfamily)[21] as positive controls, as well as the highly mobile synthetic L1 $T_F$ element L1SM[68]. As evidence of their in vitro mobility, miPSC_1_L1 and miPSC_4_L1 retrotransposed efficiently in HeLa cells, when expressed from their native promoter alone or with the addition of a cytomegalovirus promoter (Fig. 1h). Thus, endogenous L1 retrotransposition in miPSCs is driven by highly mobile donor L1s that can produce incompletely methylated, retrotransposition-competent offspring L1s.

## Single-cell miPSC clones reveal extensive L1-mediated endogenous retrotransposition

Despite de novo L1 insertions being present in 10/26 miPSC lines, we were concerned that additional mutations could be obscured by the heterogeneous mixture of cellular clones contained in bulk reprogrammed miPSCs. We therefore reprogrammed MEFs from one of our C57BL/6×129S4Sv/Jae animals (labeled I222e2), isolated individual miPSCs via FACS, and expanded 18 clones cultivated in serum until p3 (Fig. 2a). To assess the impact of standard and naïve culture conditions, respectively, upon L1 activity, each clone was divided and then further expanded in serum or 2i until p6 (Fig. 2a). We then applied ~41× average genome-wide depth Illumina WGS and mRC-seq to miPSC single-cell clones 1–9, and mRC-seq only to clones 10–18, with each clone analyzed after culture in serum or 2i media (Fig. 2a, Supplementary Fig. 1 and Supplementary Data 1). Deep WGS was performed on the parental I222e2 MEF population, attaining cumulative 117× genome-wide depth, in addition to mRC-seq (Supplementary Data 1). Using the WGS data, we again called concordant SNVs and SVs private to one miPSC clone, while excluding known germline variants and those found in the parental MEFs. We found, on average, ~100 and ~1 private SNVs and SVs per miPSC clone, respectively, almost all of which were detected in both the serum and 2i conditions for each clone (Supplementary Data 2). These frequencies resembled those found by genomic analysis of bulk miPSCs, underlining that heterogeneous and homogeneous fibroblast-derived miPSC populations are relatively free of genomic abnormalities[10,12]. This experiment also indicated choice of serum or

2i media did not impact the frequency of SNVs or SVs present in miPSCs.

By contrast, TEBreak revealed 35 putative de novo TE insertions absent from the parental MEFs, all of which were found in both serum and 2i culture conditions for at least one miPSC clone. Of these, 27 were detected by both WGS and mRC-seq, 6 by mRC-seq only, and 2 by WGS only (Supplementary Data 3). Note that the 6 events found only by mRC-seq were detected in the 9 miPSC clones (numbered 10–18) not analyzed with WGS. The 2 insertions found by WGS alone were both moderately 5′ truncated and carried a 3′ transduction, two features that reduced their probability of detection by mRC-seq, where enrichment probes target the 5′ and 3′ ends of L1 consensus sequences[28,61]. We were able to PCR amplify 32 insertions in full and capillary sequence at least their 5′ and 3′ junctions (Fig. 2b–f, Supplementary Fig. 3 and Supplementary Data 3). Two other putative TE insertions could only be amplified at their 5′ genome junction; one of these (miPSC_29_L1) however also had strong 3′ WGS and mRC-seq support. We therefore considered 33 TE insertions as validated de novo events (Table 1). Thirty-one of these were PCR validated as private to only one miPSC clone, whereas the remaining two events were found in either 2 clones (miPSC_23_B2) or 4 clones (miPSC_43_L1) (Supplementary Fig. 3). These last two insertions were therefore present in subclones of the parental MEF population.

The 33 de novo insertions included 20, 3 and 1 $T_F$, $G_F$ and A L1 subfamily members, respectively, as well as 2 B1 and 7 B2 elements (Supplementary Data 3). All insertions generated TSDs and a 3′ polyA tract, and integrated at a degenerate L1 endonuclease motif (Table 1). 14/24 L1 insertions retained at least one promoter monomer and were therefore considered full-length (Table 1). Of the remaining 10 L1s, 3 were 5′inverted. One unusual B1 insertion, miPSC_18_B1, was flanked by 5′ and 3′ polyA tracts as well as TSDs (Fig. 2d), likely arising via a variant of TPRT[69]. While no TE insertions were found in protein-coding exons, 14 were intronic, including a B2 antisense to the tumor suppressor gene *Brca1* (Fig. 2b) and an L1 $G_F$ antisense to the dystrophin gene *Dmd* (Fig. 2e). 16/18 miPSC clones (88.9%) harbored at least one de novo TE insertion, including all clones analyzed with both WGS and mRC-seq

**Table 1 | De novo TE insertions detected and PCR validated in miPSC lines**

| Insertion # | Subfamily | Location | Monomers | EN motif | TSD (bp) | PolyA (bp) | Origin | Detected by |
|---|---|---|---|---|---|---|---|---|
| miPSC_1_L1 | $T_F$ | 1q | 6.8 | TCTT/AG | 16 | ~125 | Reprogramming | WGS + mRC-seq |
| miPSC_2_L1 | $T_F$ | 10q | 0 | TTCT/GT | 14 | >100 | Mosaic | WGS + mRC-seq |
| miPSC_3_L1 | $T_F$ | 13q | 5.8 | ATTC/AA | 15 | ~50 | Reprogramming | WGS + mRC-seq |
| miPSC_4_L1 | $T_F$ | 3q | 5.3 | TCTT/AA | 13 | ~54 | Reprogramming | WGS + mRC-seq |
| miPSC_5_L1 | $G_F$ | 19q | 2 | TTAT/AT | 14 | ~50 | Reprogramming | WGS + mRC-seq |
| miPSC_6_L1 | $T_F$ | 7q | 0 | TTTA/AA | 17 | ~51 | Reprogramming | WGS + mRC-seq |
| miPSC_7_L1 | $G_F$ | Xq | 5 | TCTT/AT | 16 | >80 | Reprogramming | WGS + mRC-seq |
| miPSC_8_L1 | $T_F$ | 19q | 3.7 | TTTC/AA | 19 | ~24 | Reprogramming | WGS + mRC-seq |
| miPSC_9_B2 | B2 | 11q | NA | TCTT/AC | 16 | >60 | Reprogramming | WGS + mRC-seq |
| miPSC_10_L1 | $T_F$ | 12q | 0 | TTTT/GT | 6 | ~36* | Reprogramming | WGS |
| miPSC_11_B2 | B2 | 13q | NA | TTTT/GA | 14 | >73 | Reprogramming | WGS + mRC-seq |
| miPSC_12_L1 | $T_F$ | 13q | 0 | TCTT/AG | 17 | ~97 | Reprogramming | WGS + mRC-seq |
| miPSC_13_L1 | A | 14q | 3 | TTTC/AT | 13 | ~46 | Reprogramming | WGS + mRC-seq |
| miPSC_14_B2 | B2 | 15q | NA | TTTT/AC | 16 | >66 | Reprogramming | WGS + mRC-seq |
| miPSC_15_L1 | $G_F$ | 2q | 0 | TTTC/AA | 17 | ~28* | Reprogramming | WGS |
| miPSC_16_L1 | $T_F$ | 2q | >3 | TTTT/AA | 16 | >100 | Reprogramming | WGS + mRC-seq |
| miPSC_17_L1 | $T_F$ | 3q | >3 | ACTT/AA | 14 | ~45 | Reprogramming | WGS + mRC-seq |
| miPSC_18_B1 | B1 | 3q | NA | TTTT/AA | 15 | ~30 | Reprogramming | WGS + mRC-seq |
| miPSC_19_L1 | $T_F$ | 3q | >3 | GTTT/AT | 15 | >80 | Reprogramming | WGS + mRC-seq |
| miPSC_21_L1 | $T_F$ | 4q | 0 | TTTT/CA | 17 | >150 | Reprogramming | WGS + mRC-seq |
| miPSC_22_B2 | B2 | 6q | NA | TCTT/GA | 15 | ~52 | Reprogramming | WGS + mRC-seq |
| miPSC_23_B2 | B2 | 9q | NA | TTTT/AT | 16 | ~50 | Mosaic | WGS + mRC-seq |
| miPSC_24_B2 | B2 | Xq | NA | TTTT/AA | 15 | >100 | Reprogramming | WGS + mRC-seq |
| miPSC_26_L1 | $T_F$ | 1q | >3 | TCTT/AT | 22 | ~58 | Reprogramming | WGS + mRC-seq |
| miPSC_27_B2 | B2 | 11q | NA | TTTC/AA | 14 | >60 | Reprogramming | WGS + mRC-seq |
| miPSC_28_L1 | $T_F$ | 13q | 3.6 | TCCT/AA | 15 | ~93* | Reprogramming | mRC-seq |
| miPSC_29_L1 | $T_F$ | 15q | 0 | TCTT/AA | 16 | >80 | Reprogramming | WGS + mRC-seq |
| miPSC_30_L1 | $T_F$ | 6q | >3 | TCTT/AT | 16 | ~72 | Reprogramming | mRC-seq |
| miPSC_31_L1 | $T_F$ | 7q | >3 | TTTG/AC | 15 | ~43 | Reprogramming | mRC-seq |
| miPSC_32_L1 | $T_F$ | Xq | 2 | TCTT/AT | 13 | ~37 | Reprogramming | WGS + mRC-seq |
| miPSC_33_L1 | $G_F$ | Xq | >3 | TTTT/AA | 15 | ~47 | Reprogramming | mRC-seq |
| miPSC_34_L1 | $T_F$ | 8q | 0 | TCTT/AA | 6 | ~36* | Reprogramming | WGS + mRC-seq |
| miPSC_35_L1 | $T_F$ | 1q | 0 | TTTA/AA | 15 | ~38 | Reprogramming | WGS + mRC-seq |
| miPSC_36_L1 | $G_F$ | 8q | 0 | ATGT/GA | 6 | ~42 | Reprogramming | WGS + mRC-seq |
| miPSC_37_L1 | $T_F$ | 1q | 1.2 | TTTT/GT | 14 | ~20 | Reprogramming | WGS + mRC-seq |
| miPSC_38_L1 | $T_F$ | 10q | 0 | TTCT/AA | 15 | ~55 | Reprogramming | WGS + mRC-seq |
| miPSC_39_L1 | $T_F$ | 10q | 0 | TTTT/AA | 8 | >140* | Reprogramming | WGS + mRC-seq |
| miPSC_40_L1 | $T_F$ | 11q | >3 | TTTT/GA | 14 | >120 | Reprogramming | WGS + mRC-seq |
| miPSC_41_L1 | $T_F$ | 12q | 2.6 | TCTT/GC | 16 | ~49 | Reprogramming | mRC-seq |
| miPSC_42_B1 | B1 | 14q | NA | TTCT/AA | 15 | >50 | Reprogramming | mRC-seq |
| miPSC_43_L1 | $T_F$ | 16q | >3 | ATTT/AA | 14 | ~42* | Mosaic | WGS + mRC-seq |
| miPSC_69_L1 | $T_F$ | 13q | 2 | CTTT/AT | 16 | ~61 | Mosaic | WGS (ONT) |
| miPSC_87_L1 | $T_F$ | 17q | 0 | TTTT/GT | 16 | ~21 | Reprogramming | WGS (ONT) |

Insertions marked with an asterisk carry a 3′ transduction. miPSC_1_L1 - miPSC_8_L1 were detected in adult primary cell-derived bulk miPSCs, miPSC_9_B2 - miPSC_43_L1 in MEF-derived single-cell miPSC clones, and miPSC_69_L1 and miPSC_87_L1 in MEF-derived bulk miPSCs. Unless stated otherwise, insertions were detected by Illumina sequencing.

(Supplementary Data 1). Clone 2 contained the most (6) insertions. No de novo ERV insertions were found.

Among 277 high confidence heterozygous non-reference TE insertions (Supplementary Data 4) found in the parental MEF population, 97.0% were detected on average in each miPSC clone surveyed with WGS and mRC-seq. Down-sampling followed by seeking at least one WGS or mRC-seq read in support of these non-reference insertions suggested our approach would distinguish approximately 50%, 95 and 99% of de novo TE insertions from pre-existing subclonal TE insertions present in 1%, 5 and 10% of cells, respectively (Supplementary Fig. 4b), whereas mRC-seq alone

would achieve respective sensitivities of approximately 22%, 69 and 89%. Consistently, only 2/33 PCR validated TE insertions in the miPSC clones were subclonal in the parental MEFs (Table 1, Supplementary Fig. 3). An additional down-sampling analysis indicated de novo TE insertions were likely to be detected at a lower average WGS depth in the single-cell miPSC clones than insertions found in the bulk miPSC experiments (Supplementary Fig. 4a), in agreement with the greater homogeneity of the clonal miPSC cultivars. Deep sequencing of miPSCs and parental MEFs therefore enabled reliable detection and distinction of TE insertions arising before and during reprogramming.

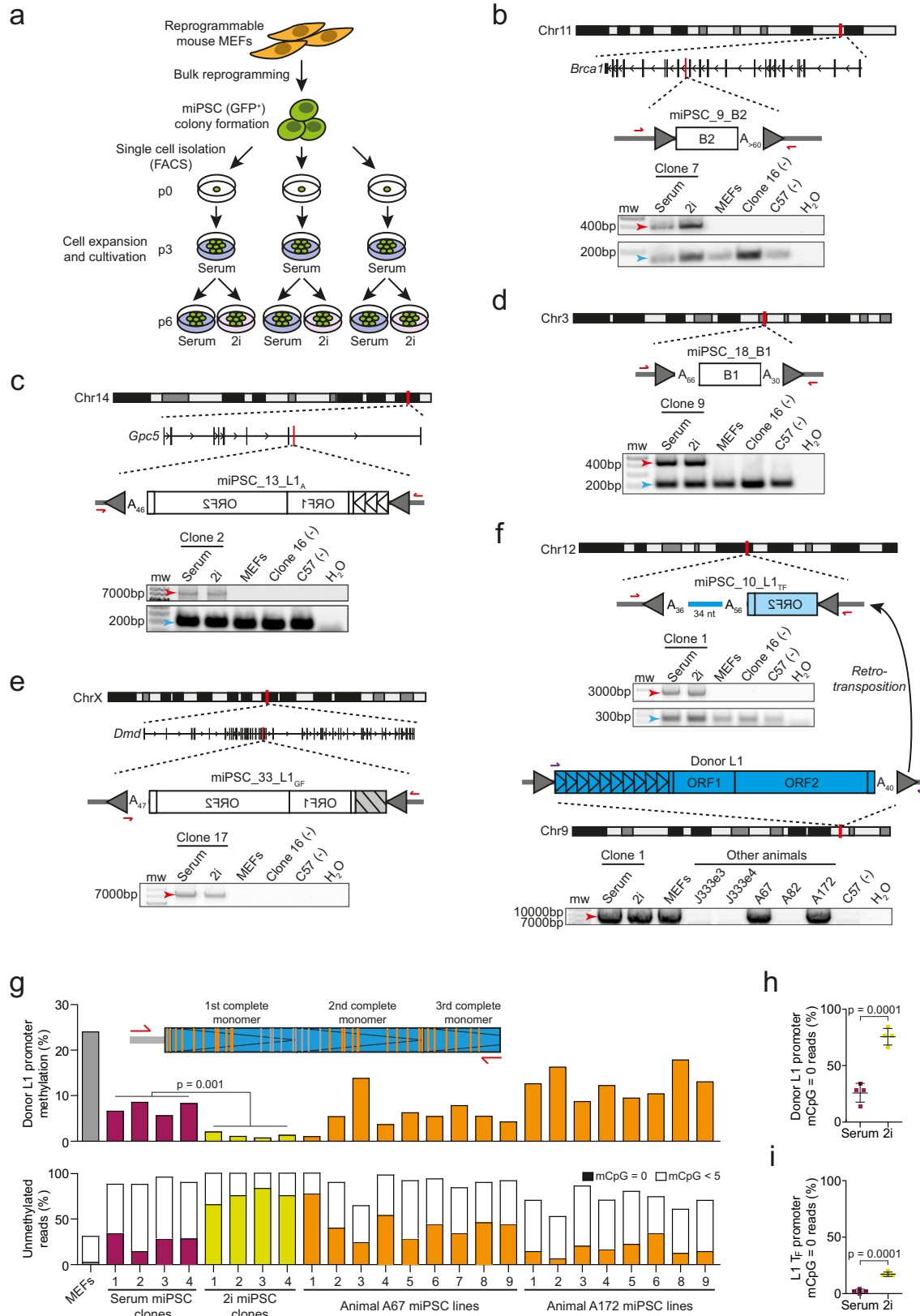

## A polymorphic retrotransposition-competent L1 eludes methylation

Six de novo L1 insertions found in the miPSC clones carried 3′ transductions, flanking sequences generated when PolII bypasses the native L1 polyA signal in favor of a downstream alternative[70–74] (Table 1). Of these transductions, 5 were either too short to reliably map to the

genome, or mapped to multiple locations (Supplementary Data 3). The remaining 34 bp transduction accompanied a 5′ truncated L1 $T_F$ insertion on Chromosome 12 (miPSC_10_L1) (Fig. 2f). While the transduction aligned uniquely to Chromosome 9, a donor L1 was not present adjacent to this reference genome location. However, PCR amplification revealed an L1 $T_F$ immediately upstream of the

**Fig. 2 | Frequent de novo TE insertions in MEF-derived clonally expanded miPSC lines. a** Experimental design to generate single-cell miPSC clonal lines. Bulk MEFs from a Col1a1-tetO-OKSM mouse (animal I222e2) were purified and reprogrammed by addition of doxycycline. Individual Oct4-GFP positive miPSCs were then isolated via FACS (p0), expanded in serum for 3 passages (p3), and then cultured in serum- or 2i-containing miPSC media for 3 additional passages (p6). DNA was then extracted and analyzed by WGS and mRC-seq for 9 single-cell clones (for both serum and 2i conditions), with 9 further clones analyzed only with mRC-seq. **b** A full-length de novo B2 inserted and orientated in antisense to intron 15 of *Brca1*. PolyA tract length is indicated immediately 3′ of the B2. TSDs are depicted as gray arrows flanking the B2. PCR validation (gel pictures shown) involved an empty/filled PCR design where both primers (red arrows) are outside of the B2, generating filled B2 (red arrow) and empty wild-type (blue arrow) products. The B2 was amplified only in either the serum or 2i conditions for the single-cell clone (number 7) where the B2 was detected by genomic analysis, and not in the matched parental MEFs, the C57BL/6 strain, or a single-cell clone (number 16) selected at random. Molecular weight (mw) markers are provided at the left of gel images. **c** A full-length (3 monomers) L1 A subfamily element inserted de novo antisense to intron 7 of *Gpc5*. Sequence characteristics and PCR validation results are shown as in panel (**b**). Promoter monomers are shown as triangles within the L1 5′UTR. **d** As in (**b**), except showing an unusual intergenic B1 insertion flanked by both 5′ and 3′ polyA tracts. **e** A full-length L1 $G_F$ inserted de novo antisense to intron 60 of *Dmd*. PCR validation involved a 5′ genomic primer and a 3′ junction primer (red arrows). As indicated by

a gray box with black stripes, the number of monomers is unknown but was >3. **f** A heavily 5′ truncated, intergenic de novo L1 $T_F$ insertion validated by empty/filled PCR, as per (**b**). Sequence features are annotated as per (**b**), with the addition of a 34nt 3′ transduction matching a donor L1 $T_F$ located on Chromosome 9. PCR using primers (purple arrows) designed to amplify the entire donor L1 indicated it was polymorphic in our colony. Capillary sequencing indicated the donor L1 retained a promoter of 10 monomers and had intact ORFs. **g** Locus-specific bisulfite sequencing analysis of the donor L1 promoter identified in panel (**f**), in MEFs, single-cell miPSC clones, and miPSC lines derived from primary cells. top: Assay design and primer locations. CpGs located in the first 3 monomers of the donor L1 were assessed. Orange and gray strokes indicate CpGs covered and not covered, respectively, by sequencing the amplicon with 2×300mer Illumina reads. middle: Mean percentages of donor L1 CpG methylation for 50 non-identical sequences selected at random from each sample. A two-tailed t test was used to compare serum and 2i culture conditions for single-cell miPSC clones 1–4. bottom: Percentages of fully unmethylated (mCpG = 0, filled bars) and heavily unmethylated (0 < mCpG < 5, white bars) reads using the same sequencing data as displayed in the above histogram. **h** Percentages of fully unmethylated (mCpG = 0) reads corresponding to the donor L1 promoter identified in panel (**f**), for miPSCs cultured in serum or 2i conditions. Data represent mean methylation ± SD observed for single-cell miPSC clones 1–4 (n = 4 biological replicates of each condition). Significance testing was via two-tailed t test. **i** As for (**h**), except using an assay targeting the L1 $T_F$ subfamily monomer. Source data are provided as a Source Data file.

transduced sequence (Fig. 2f). This donor L1 was polymorphic in our C57BL/6 × 129S4Sv/Jae animals and retained a 5′ promoter comprising an unusually high number of monomers (10). Capillary sequencing confirmed the donor L1 possessed intact ORFs. L1 locus-specific bisulfite sequencing revealed that few (24.1%) of the CpG dinucleotides in the first two monomers of the donor L1 promoter were methylated in MEFs (Fig. 2g and Supplementary Fig. 5), as opposed to 7.3% in a subset of single-cell miPSC clones cultured in serum, and 1.3% for the same miPSC clones when cultured in 2i conditions (Fig. 2g). This difference in CpG methylation between culture conditions was significant ($p = 0.001$, two-tailed t test). The donor L1 promoter was fully unmethylated in nearly all miPSCs cultured in 2i (Fig. 2g and Supplementary Fig. 5). Indeed, significantly more ($p = 0.0001$, two-tailed t test) fully unmethylated sequences were found for the donor L1 promoter in 2i conditions than in serum, possibly as a consequence of global naïve state hypomethylation (Fig. 2h). Among the bulk reprogrammed miPSCs obtained from animals A67 and A172, which carried the donor L1 (Fig. 2f), only 9.1% of CpG dinucleotides were methylated in the donor L1 promoter, and fully unmethylated sequences were identified in all miPSC lines (Fig. 2g and Supplementary Fig. 5). By contrast, in MEFs, 83.6% of CpG dinucleotides in L1 $T_F$ promoter monomers genome-wide were methylated, compared to 45.2% among the A67 and A172 miPSC lines (Supplementary Fig. 6), and resembled L1 $T_F$ promoter methylation reported elsewhere for differentiated primary cells[48,61]. L1 $T_F$ subfamily monomers were also significantly ($p = 0.001$, two-tailed t test) less methylated in 2i (34.3%) miPSC conditions than serum (53.5%), leading to an increase in fully unmethylated monomers (Fig. 2i and Supplementary Fig. 6). These bisulfite sequencing analyses highlighted genome-wide and persistent relaxation of L1 $T_F$ methylation in miPSCs, leaving mobile L1 promoters completely unmethylated.

## Reprogramming is unaffected by L1 reverse transcriptase inhibition

Lamivudine (3TC) is a potent nucleoside reverse transcriptase inhibitor known to limit engineered L1 retrotransposition without impacting telomerase or ERV mobility[75,76]. In previous retrotransposition assays conducted in cultured HeLa cells, 3TC was tested at a maximum concentration of 25 μM against the codon-optimized L1SM element, reducing its mobility by ~50%[75]. By performing titration experiments to optimize the use of 3TC during miPSC generation, we determined that 3TC concentrations of up to 100 μM did not reduce viability of cultured MEFs or miPSCs (Supplementary Fig. 7a) or MEF

reprogramming efficiency (Fig. 3a and Supplementary Fig. 7b–e). Using a wild-type L1 $T_F$ carrying an mCherry retrotransposition indicator cassette (L1-mCherry), we found 100 μM 3TC reduced mouse L1 retrotransposition by ~95% in cultured HeLa cells (Fig. 3b and Supplementary Fig. 8). Thus, 3TC could be used, without apparent drawbacks, to limit L1-mediated mutagenesis arising during reprogramming and miPSC cultivation.

## Nanopore genomic analysis of TE insertions in bulk miPSCs

A single DNA sequencing read, if of sufficient length and quality, can completely resolve a de novo TE insertion present in a heterogeneous cell population, as well its genomic flanks and accompanying TPRT hallmarks[77]. Long-read sequencing, as for example developed by Oxford Nanopore Technologies (ONT), is well suited to this application, and can locate TE insertions within repetitive genomic regions refractory to short-read methods[35,77–79]. As a proof-of-principle before analyzing miPSCs, we applied PCR-free ONT sequencing to 5 HeLa cell lines expanded from single colonies (~5× genome-wide depth per colony) harboring mouse L1-mCherry retrotransposition events (Supplementary Data 1). Using the TLDR long-read TE insertion detection pipeline[35], we identified 41 L1-mCherry insertions spanned by at least one ONT read (Supplementary Data 3). L1-mCherry insertions overwhelmingly bore TPRT hallmarks regardless of whether they were detected by one, or more than one, ONT read (Fig. 3c), showing that single spanning ONT reads can reliably recover bona fide retrotransposition events.

Next, to survey endogenous retrotransposition in miPSCs, we ONT sequenced (~20× average genome-wide depth) 6 bulk miPSC lines reprogrammed in the presence of 100 μM 3TC, 6 control miPSC lines not treated with 3TC, and matched parental MEFs (Fig. 3a and Supplementary Data 1). Using these data, TLDR identified 3,975 non-reference TE insertions carried by the parental MEFs (Supplementary Data 4). Of these, 3,429 (86.3%) corresponded to previously known insertions[56] and 99.6% were found in each miPSC line, on average (Supplementary Data 4). To gauge the general tractability of PCR validation applied to this dataset, we used a panel of 4 heterozygous non-reference TE insertions (Supplementary Data 3) and confirmed these all PCR amplified in the MEFs and miPSCs (Supplementary Fig. 9a).

An additional 43 TE insertions were each detected in only one miPSC line and not the parental MEFs, or in the earlier Illumina sequencing, and were supported by at least one spanning ONT read

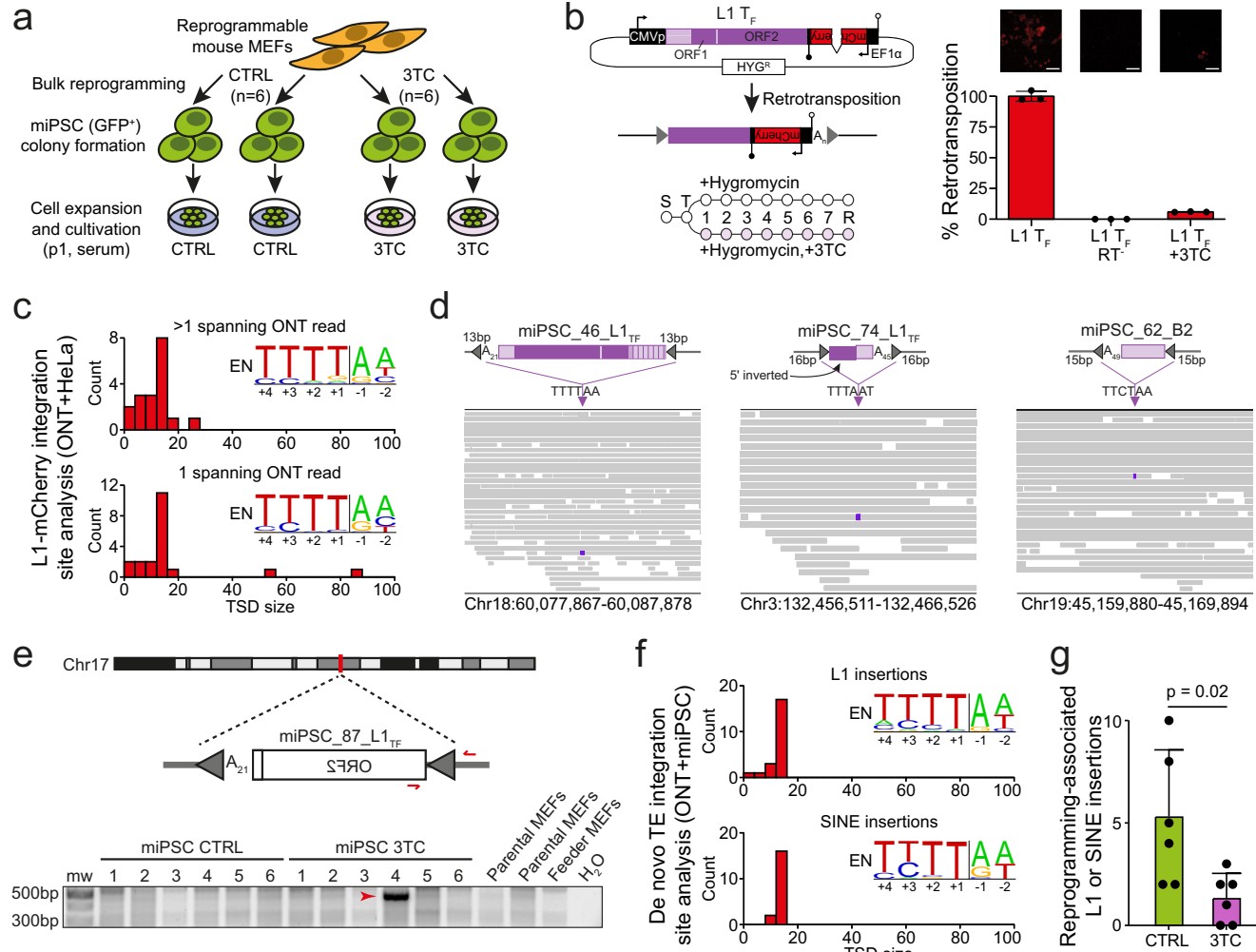

**Fig. 3 | Long-read detection of retrotransposition in HeLa cells and MEF-derived bulk miPSC lines. a** Bulk MEFs were reprogrammed by the addition of doxycycline. *Oct4*-GFP positive miPSCs were then sorted and expanded in serum. Six miPSC lines were reprogrammed and cultured in media containing 100 μM lamivudine (3TC), and six miPSC lines generated without lamivudine (CTRL). DNA was extracted from MEFs and miPSCs and then ONT sequenced. **b** Top left: retrotransposition indicator plasmid L1-mCherry consists of the pCEP4 backbone (CMV promoter, black; SV40 polyadenylation signal, open lollipop; hygromycin resistance gene HYG$^R$, white) containing a wild-type L1 T$_F$ element (5′UTR, light purple; ORFs, dark purple). An mCherry reporter gene equipped with an EF1α promoter and HSVtk polyadenylation signal (black lollipop) is inserted into the L1 3′ UTR antisense to the L1. The mCherry sequence is interrupted by an intron in sense orientation relative to the L1, ensuring mCherry expression only upon retrotransposition. bottom left: retrotransposition assay timeline. Cells were split (S), transfected (T), and cultured in hygromycin-containing medium with and without 100 μM 3TC. Retrotransposition efficiency was assessed by flow cytometry 8 days post-transfection (R). top right: fluorescence microscopy images showing representative wells at 8 days post-transfection with L1 T$_F$ (left), reverse transcriptase mutant (RT⁻) L1 T$_F$ (middle), and L1 T$_F$ treated with 100 μM 3TC (right). Scale bars (white) represent 100 μm. bottom right: Retrotransposition efficiency assessed by flow cytometry, relative to L1 T$_F$. Histogram depicts the mean ± SD of three independent biological replicates (black dots, *n* = 3) consisting of three technical replicates each. **c** Target site duplication (TSD) size distributions for L1-mCherry retrotransposition events recovered from HeLa cells via TLDR analysis of ONT sequencing, divided into integrants detected by >1 (top) or 1 (bottom) spanning read. Inset sequence logos[120] display the observed integration site motif, as preferred by the L1 endonuclease (EN). **d** Integrative genomics viewer (IGV)[121] inspection of 3 example de novo TE insertions. Cartoons (top) show a full-length (>6 monomers) L1 T$_F$ insertion, a heavily 5′ truncated L1 T$_F$ insertion, and a SINE B2 insertion, each flanked by TSDs (gray triangles) and a 3′ polyA tract, and integrated at the indicated L1 EN motif. IGV snapshots (bottom) show ONT read alignments from the miPSC line carrying each TE insertion (purple box). **e** A 5′ truncated intergenic de novo L1 T$_F$ insertion. PolyA (A$_n$) tract length is indicated immediately 3′ of the L1. PCR validation primers are shown as red arrows. Underneath is shown a PCR validation agarose gel containing the 5′ junction PCR product (red arrow) only in the miPSC line where the L1 was detected by ONT sequencing. Molecular weight (mw) markers are provided at the left of the gel. Note: DNA was obtained from two parental MEF aliquots, one corresponding to miPSC CTRL/3TC lines 1–3 and one to miPSC CTRL/3TC lines 4–6. **f** TSD size distributions for de novo L1 (top) and SINE (bottom) insertions detected in miPSC lines via ONT sequencing, with inset integration site sequence logo, as per (**c**). **g** Putative reprogramming-associated L1-mediated insertion counts detected by ONT sequencing in CTRL and 3TC-treated miPSCs. Replicate (*n* = 6 per group) data points are marked by black dots and represented as the mean ± SD. Significance was calculated via a two-tailed *t* test. Source data are provided as a Source Data file.

(Fig. 3d and Supplementary Data 3). Performing PCR validation of these insertions, a step mainly intended to exclude them being polymorphisms carried by the parental or feeder MEF populations, we could amplify one (miPSC_50_B1) in the parental MEFs (Supplementary Fig. 9b). The remaining 42 putative de novo events comprised 16, 4 and

2 L1 T$_F$, G$_F$, and A insertions, respectively, as well as 5 and 13 SINE B1 and B2 insertions, respectively, and 2 ERV insertions. One L1 T$_F$ insertion (miPSC_69_L1) PCR amplified in multiple miPSC lines (Supplementary Fig. 9c) and one L1 T$_F$ insertion (miPSC_87_L1) amplified only in the miPSC line where it was detected by ONT sequencing (Fig. 3e). While

the remaining insertions could not be PCR amplified in any sample, all 40 of the putative de novo L1 and SINE insertions carried clear TPRT hallmarks (Fig. 3f and Supplementary Data 3) and on this basis we considered them bona fide retrotransposition events. As well, both ERVs corresponded to the mobile intracisternal A-particle (IAP) family, presented a typical proviral structure of two long terminal repeats flanking an internal coding sequence, and generated TSDs of the expected size (6 bp)[80,81] (Supplementary Data 3). It was not clear whether the difference in (two) de novo ERV insertions being found here by ONT sequencing, and none by the earlier Illumina sequencing of different miPSC lines, was due to chance or unknown technical reasons. Significantly fewer ($p = 0.02$, two-tailed t test) putative reprogramming-associated L1-mediated insertions were found on average in the 3TC-treated miPSCs (-1.3 per line) than in the control miPSCs (-5.2 per line) (Fig. 3g), consistent with L1 inhibition by 3TC (Fig. 3b). Overall, detection of endogenous retrotransposition events in bulk miPSCs by ONT sequencing yielded results orthogonal and complementary to our short-read genomic analyses.

### Genome-wide DNA demethylation during reprogramming focused on young L1 loci

A major feature of reprogramming mouse fibroblasts to a pluripotent state is globally reduced DNA methylation[43,44,47,48]. Although bisulfite sequencing can estimate the overall methylation of TE families, at specific genomic loci it can typically only resolve CpGs close to the termini of full-length L1s not located in highly repetitive regions. To generate a comprehensive genome-wide view of DNA methylation changes during reprogramming, and complement our bisulfite sequencing data, we analyzed the ONT data obtained from MEFs and miPSCs (control and 3TC-treated) using Methylartist[35,82]. While methylation in control miPSCs was reduced genome-wide when compared to MEFs, on protein-coding gene promoters and amongst all of the TE families considered, the very youngest L1 subfamilies ($T_{FI}$ and $T_{FII}$) displayed by far the greatest median methylation change (−54.0%) (Fig. 4a, Supplementary Fig. 10a, b and Supplementary Data 5). 90.3% of full-length L1 $T_{FI}$ and $T_{FII}$ copies were significantly ($p < 0.01$, Fisher's exact test with Bonferroni correction, and ΔmC >25%) less methylated in control miPSCs than MEFs (Supplementary Data 5), with this demethylation most pronounced in the monomeric L1 5′UTR (Fig. 4b). By contrast, we observed no significant local or global differences in L1 $T_F$ or protein-coding gene promoter methylation between control and 3TC-treated miPSCs (Fig. 4 and Supplementary Data 5). Thirty-six L1s initiated transcription of a spliced mRNA from their 5′UTR, as defined by GenBank expressed sequence tags, including alternative promoters for protein-coding genes expressed in pluripotent cells, such as *Fsd1l* (Fig. 4c and Supplementary Data 5). We also identified full-length L1s demethylated in both MEFs and miPSCs (Supplementary Fig. 10c), in line with prior human data suggesting certain L1 loci evade DNA methylation in differentiated cells[34,83]. In sum, ONT analysis showed global reprogramming-associated demethylation was most accentuated for the youngest L1s, where retrotransposition potential is concentrated, creating opportunities for L1-driven mobilization and protein-coding gene alternative promoters.

## Discussion

This study demonstrates miPSCs incompletely silence mobile TE families and routinely harbor de novo TE insertions. While some TE insertions occur in parental cells and are inherited by miPSCs, our data suggest the majority arise during reprogramming or very early upon reaching pluripotency. In support of this view, firstly, we observed profound hypomethylation of young L1 promoters in miPSCs and not parental cells. As shown elsewhere, L1 mRNA abundance is low in fibroblasts and increases greatly upon reprogramming[45,46,52], while engineered L1 reporter genes retrotranspose >10-fold more frequently in hiPSCs and human ESCs than in fibroblasts[51,52]. Secondly, 38/41 de

novo TE insertions detected by Illumina sequencing PCR validated in only one miPSC line each. These and the 42 putative de novo TE insertions identified by ONT sequencing were absent from all other samples in the study, as assayed by PCR and deep WGS. Finally, given the similar mutation calling thresholds of Illumina sequencing applied to heterogeneous (bulk) and homogenous (single-cell clone) miPSC populations, we note that private SNVs (-100 per line) and SVs (-1 per line) were detected at similar frequencies in either experiment, whereas far more de novo retrotransposition events were found in the latter dataset. One explanation for this result is that a relatively small number of clones dominate bulk reprogramming experiments[84] and most SNVs and SVs predate reprogramming[10,11,13], while retrotransposons mainly mobilize during reprogramming. This position is consistent with a prior WGS analysis that, alongside thousands of SNVs, identified no somatic L1 insertions among 10 human fibroblast clones generated from single cells[85]. It is nonetheless possible that additional somatic variants would have been annotated if primary single-cell clones, as analyzed elsewhere[86], were reprogrammed. However, the introduction of multiple bottlenecks followed by clonal expansion prolongs cell culture and could thereby exaggerate mutation frequencies. The parental MEFs used here cannot in any case be clonally expanded from single cells, and reprogram extremely inefficiently after more than three passages in culture[87,88], and for these reasons we did not prepare single-cell MEF clones prior to reprogramming.

Previous experiments employing hiPSCs and mouse and human ESCs showed L1 de-repression and mobilization were likely to take place in pluripotent cells[34,41,42,45,46,48,50–52,89]. Notably, 31/56 (55.4%) de novo L1 insertions found here in miPSCs were full-length, a similar percentage to that observed previously in hiPSCs (57.1%)[45]. New full-length L1 insertions have potential for further retrotransposition and were largely unmethylated in miPSCs. Their CpG dinucleotides presented a "sloping shore" of methylation, as found elsewhere for newly retrotransposed CpG islands[34,35,48,90], where methylation decreases from the L1 5′ genome junction and forms a trough before sharply increasing over the L1 ORFs. Only 3/56 (5.4%) L1 insertions corresponded to the A subfamily, while the remainder were $T_F$ and $G_F$ elements, consistent with relative activity levels revealed by sequencing extended mouse pedigrees and mouse tumors[28,61]. De novo SINE B1 and B2 insertions, mediated in trans by the L1 protein machinery[91], were also detected in miPSCs, in line with L1-mediated *Alu* SINE insertions arising in hiPSCs and human ESCs[45,89,92]. Discovery of de novo TE insertions in low-passage miPSCs derived from multiple parental cell types suggests endogenous retrotransposition may be an intrinsic risk of the epigenome remodeling required for the acquisition of pluripotency[7,43–45,47].

Exonic retrotransposon insertions are clear in their potential to cause disease[93]. However, the L1 endonuclease does not favor exons[65,66], which are depleted for its AT-rich recognition motif and make up only ~2% of the genome[94], and none of the 81 de novo L1-mediated insertions reported here in miPSCs were exonic. By comparison, L1 integration within introns occurs much more frequently. We observed 30/81 (37.0%) L1-mediated insertions in introns, in line with the proportion of the genome occupied by these regions[65,66]. Whilst intronic events are less likely to be pathogenic than exonic insertions, they can perturb gene expression, for example through reduced transcriptional elongation[95] or, as shown here, the provision of new promoter elements[96]. Retrotransposition into the introns of protein-coding genes, as observed here for *Brca1* and *Dmd*, could therefore undermine miPSC models of human disease. Such mutations necessitate screening of miPSC lines[4]. Fortunately, strategies to minimize TE-mediated mutagenesis, including via the use of 3TC or another L1 reverse transcriptase inhibitor[75], appear achievable without affecting DNA methylation or reprogramming efficiency. While the frequency of de novo endogenous L1-mediated insertions found in

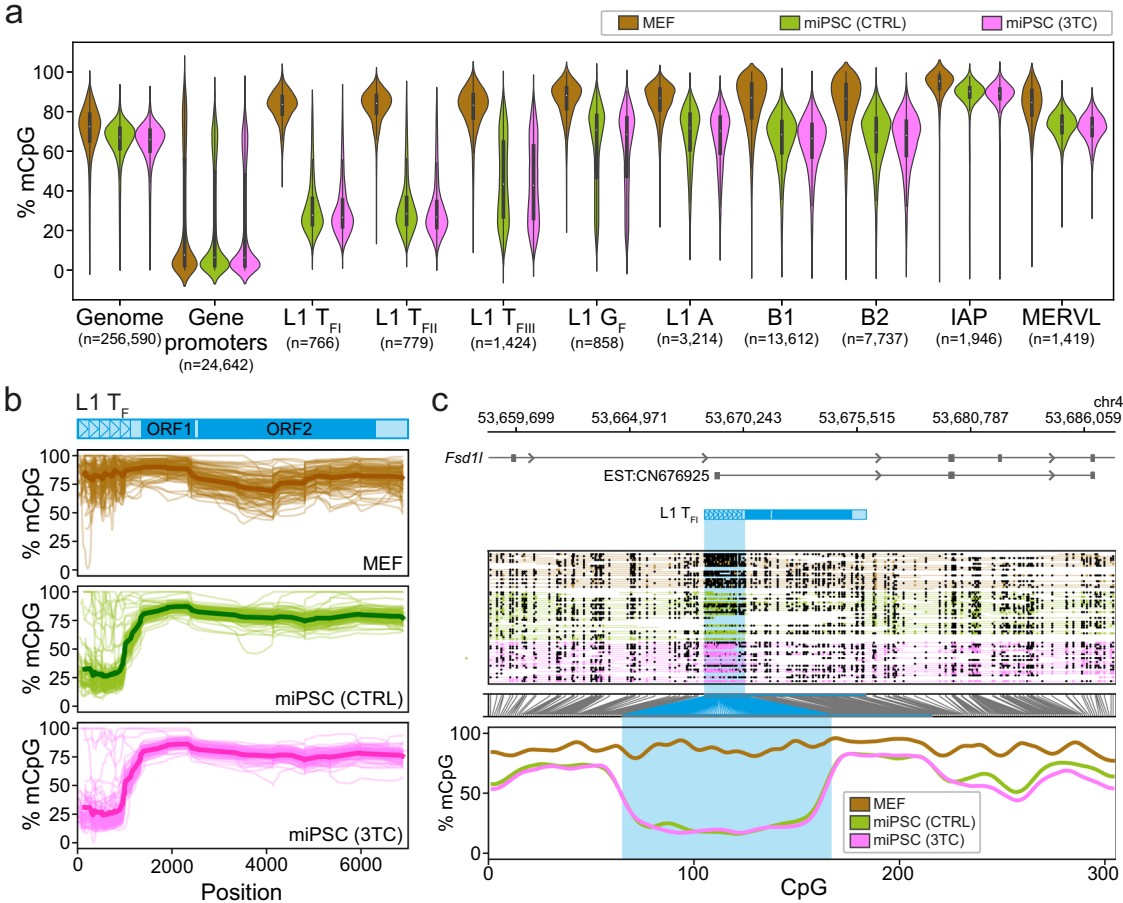

**Fig. 4 | Genome-wide methylation analyses via ONT sequencing. a** CpG methylation ascertained by ONT sequencing of MEFs (brown), control miPSCs (CTRL, green), and miPSCs derived in the presence of lamivudine (3TC, pink). Results are shown for the whole genome (10kbp windows), the proximal promoters (−1000, +500) of protein-coding genes[119], the 5′UTR of $T_F$, $G_F$, and A-type L1s > 6kbp, B1 and B2 SINEs, and MERVL MT2 and IAP long terminal repeats. Note: the 6 control and 3TC-treated miPSC replicates are displayed in aggregate to compare against the single MEF replicate, and the number of elements in each category is shown underneath. Box plots indicate the median, interquartile range, and 1.5× interquartile range. **b** Composite L1 $T_F$ methylation profiles. Each graph displays 100 profiles. A schematic of the $T_F$ consensus is provided at top. Average values are indicated by more thickly colored lines. **c,** Methylation profile of the *Fsd1l* locus obtained by ONT sequencing. The first panel shows an L1 $T_F$ orientated in a sense to intron 6 of *Fsd1l*, as well as an expressed sequence tag (EST) obtained from a mouse ESC sample and supporting a transcript initiated in the $T_F$ 5′UTR and spliced into a downstream *Fsd1l* exon. The second panel displays ONT read alignments, with unmethylated CpGs colored in brown (MEFs), green (control miPSCs) and pink (3TC-treated miPSCs), methylated CpGs colored black, and CpGs not confidently called, i.e. abs(log-likelihood ratio) > 2.5, omitted. The displayed miPSC data were downsampled to the approximate depth of the MEF data with Picard DownsampleSam version 2.27.4. The third panel indicates the relationship between CpG positions in genome space and CpG space, including those corresponding to the $T_F$ 5′UTR (shaded light blue). The fourth panel indicates the fraction of methylated CpGs. Note: this L1 $T_F$ is polymorphic amongst inbred mouse strains[56]. Source data are provided as a Source Data file.

miPSCs was significantly reduced by 3TC, a handful of events, likely arising prior to reprogramming, were nonetheless observed in 3TC-treated miPSCs. Reverse transcriptase inhibitors thus may be incorporated into future miPSC derivation protocols to attenuate de novo retrotransposition, in addition to genomic screening of miPSC lines.

## Methods

### Ethics statement
All animal experimentation was performed under the auspices and approval of the Monash University Animal Research Platform Animal Ethics Committee (Approval Numbers MARP-2011-172-Polo, MARP-2011-171-BC-Polo, MARP-2017-151-BC-Polo, and ERM# 21634).

### Adult *Oct4*GFP-OKSM-M2rtTA mouse tissue somatic cell isolation and reprogramming
Induced pluripotent stem cells were generated from adult (females aged 39–57 days) and embryonic (E13.5) *Oct4*GFP-OKSM-M2rtTA doxycycline inducible reprogrammable mice[53]. These animals are heterozygous for an *Oct4*-GFP reporter and an OKSM cassette targeted to the *Collagen1α1* locus, and homozygous for the ROSA26-M2rtTA allele from the ubiquitous *ROSA26* locus. The polycistronic cassette is under the control of a tetracycline-dependent promoter *(tetOP)*. Hence, upon the addition of doxycycline, M2rtTA binds to the *tetOP*, thereby inducing OKSM expression. *Oct4*GFP-OKSM-M2rtTA mice were housed at the Monash University Animal Research Platform animal facility. Animals were kept under standard housing conditions, with a 12/12 h dark/light cycle, ambient room temperature of 18–24 °C, and 40–70% humidity.

Bone marrow extraction and FACS purification of granulocytes and hematopoietic stem (LSK) cells were performed as previously described[97]. In brief, harvested bone marrow cells were labeled using a two-step sequential antibody labeling procedure using the following primary conjugated antibodies: 1:200 dilution of Anti-Mouse CD5 FITC antibody (BD Biosciences, Cat#: 553020), 1:100 dilution of Anti-Mouse B220 FITC antibody (BD Biosciences, Cat#: 557669), 1:200 dilution of Anti-Mouse TER-119 FITC antibody (BD Biosciences, Cat#: 557915),

1:400 dilution of Anti-Mouse Sca-1 PB antibody (Biolegend, Cat#: 122519), 1:200 dilution of Anti-Mouse cKit APC antibody (BD Biosciences, Cat#: 553356), 1:200 dilution of Anti-Mouse SSEA1 Biotinylated antibody (Thermo Fisher Scientific, Cat#: 13-8813-82), 1:200 dilution of Anti-Mouse Gr-1 APC-Cy7 antibody (Biolegend, Cat#: 108423) and 1:1000 dilution of Anti-Mouse Mac1 PE antibody (Biolegend, Cat#: 101207). This was followed by the secondary labeling step with 1:200 dilution of Streptavidin PE-Cy7 antibody (BD Biosciences, Cat#: 557598). Cells were isolated and sorted using an Influx Cell Sorter Instrument (BD Biosciences) with a 100 μm nozzle. Samples were resuspended in phosphate buffered saline (PBS) supplemented with 2% fetal bovine serum (Thermo Fisher Scientific, Cat#: SH30071.03FBS, Hyclone). FACS sorting for these and the cell types below were performed with 2 μg/mL Propidium Iodide (PI) (Sigma Aldrich, Cat#: P4864) in order to exclude non-viable cells. Granulocytes were isolated using the following cell surface marker profile: $CD5^-/B220^-/Ter119^-/Sca1^-/cKit^-/SSEA1^-/Gr1^+/Mac1^+$, whilst LSK cells were isolated from bone marrow using the following cell surface marker profile: $CD5^-/B220^-/Ter119^-/Sca1^+/cKit^+/SSEA1^-/Gr1^-/Mac1^-$.

Fibroblasts were isolated from both ear lobes from each mouse. Tissue pieces were resuspended in 0.25% Trypsin-EDTA (Thermo Fisher Scientific, Cat#: 25200-072) solution, and after 5 min incubation at room temperature, were mechanically minced using two surgical blades for a further 2 min. iPSC medium was used to inactivate trypsin, and dissociated pieces were transferred to a 15 mL centrifuge tube (Corning). Tissue pieces were then transferred to a gelatin coated T-75 flask (Corning) and cells were left to grow for a further 7 days. $CD45^-/CD31^-/Thy1.2^{hi+}$ fibroblasts were fractionated by FACs using the following antibodies: a 1:100 dilution of Anti-Mouse CD31 antibody conjugated to FITC (Thermo Fisher Scientific, Cat#: 11-0311-81), a 1:100 dilution of Anti-Mouse CD45 antibody conjugated to FITC (Thermo Fisher Scientific, Cat#: 11-0451-810) and a 1:400 dilution of Anti-Mouse Thy-1.2 antibody conjugated to APC (Thermo Fisher Scientific, Cat#: 17-0902-81).

Liver epithelial cells were isolated according to an adaptation of a previously described method[98]. Briefly, 3 mg/mL Collagenase Type 1 (Sigma-Aldrich, Cat#: C1639) solution was prepared in sterile PBS. Whole liver was transferred into a sterile 6 cm petri dish and finely minced using fine dissecting scissors. Minced liver pieces were transferred to 15 mL tube with preheated Collagenase Type 1 (Sigma, Cat#: C1639). Tubes were left to agitate on a Thermomix (Eppendorf) at 750 rpm, 37 °C for 15 min. Following digestion, the tube was removed and the cellular suspension was triturated with an 18 G needle, until tissue chunks were mostly dissociated. Sample tubes were then left to agitate for an additional 15 min, until liver fragments were completely digested. The sample suspension was again triturated, with a 21 G needle, to generate a single cell suspension, and then processed through a 40 μm cell strainer into a clean 50 mL centrifuge tube (Corning). After rinsing in 2% FCS/PBS (wash buffer) and centrifuging for 5 min at 1380 rpm for 4 °C, the supernatant was removed and cells were resuspended in wash buffer and centrifuged once again. Cells were counted and $5 \times 10^6$ cells were resuspended for sorting. Cells were labeled with primary antibodies using a 1:100 dilution of Anti-mouse CD31 antibody conjugated to FITC (Thermo Fisher Scientific, Cat#: 11-0311-81), followed by a 1:100 dilution of Anti-mouse CD45 antibody conjugated to FITC (Thermo Fisher Scientific, Cat#: 11-0451-81) and 1:100 dilution of Anti-mouse EpCAM antibody conjugated to eFluor450 (Thermo Fisher Scientific, Cat#: 48-5791-82). Liver epithelial cells were isolated using the following cell surface marker profile: $CD45^-/CD31^-/EpCAM^{+hi}$.

Thymus tissue was processed for thymic epithelial cell isolation as previously described[99]. Cells were labeled with the following antibodies: 1:400 dilution of Anti-mouse CD45 antibody conjugated to APC-Cy7 (BD Biosciences, Cat#: 557659), 1:200 dilution of Anti-mouse TER-119 antibody conjugated to APC-Cy7 (BD Biosciences, Cat#: 560509), 1:6000 dilution of Anti-mouse MHC Class II antibody conjugated to PB (Biolegend, Cat#: 107620) and 1:1000 dilution of Anti-mouse EpCAM antibody conjugated to APC (Biolegend, Cat#: 118214). Thymic epithelial cells were sorted according to the following cell surface marker profile: $CD45^-/Ter119^-/MHC$ Class $II^+/EpCAM^+$.

Intestinal stem cells were purified as previously described[100]. Cells were labeled with a 1:200 dilution of Anti-mouse CD45 antibody conjugated to BV510 (BD Biosciences, Cat#: 563891), 1:200 dilution of Anti-mouse CD31 antibody conjugated to BV510 (BD Biosciences, Cat#: 563089), a 1:100 dilution of Anti-mouse CD24 antibody conjugated to Pe-Cy7 (Thermo Fisher Scientific, Cat#: 25-0242-82), a 1:100 dilution of Anti-mouse EpCAM antibody conjugated to eFluor450 (Thermo Fisher Scientific, Cat#: 48-5791-82), and 1:100 Anti-EphrinB2 unconjugated antibody (BD Biosciences, Cat#: 743763). In the secondary labeling step, a 1:200 dilution of Anti-mouse Alexa Fluor 555 polyclonal antibody (Thermo Fisher Scientific, Cat#: A-31570) was used to detect the EphrinB2 antibody. Intestinal stem cells were fractionated according to the following cell surface marker profile: $CD45^-/CD31^-/CD24^+/EpCAM^+/Ephrin^+$.

To obtain astrocytes, brain tissue was processed using a MACS Neural Tissue Dissociation Kit (T) (Miltenyi Biotec, Cat#: 130-093-231) and manually dissected according to manufacturer's instructions. Cells were then collected and incubated with antibodies directed against Glast1 (Allophycocyanin-conjugated, ACSA-1, 1:10 dilution) (Miltenyi Biotec, Cat#: 130-098-803), 1:100 dilution of Anti-mouse CD133 antibody conjugated to PE (Thermo Fisher Scientific, Cat#: 12-1331-80), 1:200 dilution of Anti-mouse CD45 antibody conjugated to PE-Cy7 (BD Biosciences, Cat#: 552848) and 1:200 dilution of Anti-mouse CD31 antibody conjugated to PE-Cy7 (Thermo Fisher Scientific, Cat#: 25-0311-82). Astrocytes were sorted and purified according to the following cell surface marker profile: $CD45^-/CD31^-/CD133^-/GLAST1^+$.

Keratinocytes and bulge stem cells were isolated from epidermis as previously described[101]. Cells were collected and incubated with antibodies against Anti-Mouse Integrin alpha 6 antibody (GoH3) conjugated to PE (1:600) (Abcam, Cat#: ab95703), a 1:200 CD104 antibody conjugated to FITC (Biolegend, Cat#: 123605) and a 1:100 dilution of Anti-mouse CD34 biotinylated antibody (Thermo Fisher Scientific, Cat#: 13-0341-85) for 20 min at 4 °C. For secondary antibody labeling, cells were incubated with 1:200 APC-Streptavidin antibody (Biolegend, Cat#: 405207) to detect CD34 biotinylated antibody for 20 min at 4 °C. They were then washed and resuspended in PI (2 μg/mL) 1% BSA/PBS (Sigma-Aldrich, Cat#: A8412) and passed through a 40 μm cell strainer (BD Falcon) to produce single cell suspensions. Cells with the surface marker profile of $CD104^+/CD34^+/\alpha6\text{-integrin}^+$ were defined as bulge stem cells, and those marked as $\alpha6\text{-integrin}^-/CD34^+$ were defined as keratinocytes.

Reprogramming of the above 9 primary cell types was performed as follows: cells were seeded into gelatinized tissue culture treated 6-well plates (Corning Costar, Cat#: CLS3506) and cultured at 37 °C and 5% $CO_2$ in iPSC media containing KnockOut DMEM (Thermo Fisher Scientific, Cat#: 10829-018), 15% Fetal Bovine Serum (FBS) (Thermo Fisher Scientific, Cat#: SH30071.03), GlutaMAX Supplement (Thermo Fisher Scientific, Cat#: 35050061), Penicillin-Streptomycin (Thermo Fisher Scientific, Cat#: 15070063), MEM Non-Essential Amino Acids Solution (Thermo Fisher Scientific, Cat#: 11140050), 2-Mercaptoethanol (Thermo Fisher Scientific, Cat#: 21985023) and 1000 U/mL Leukemia Inhibitory Factor (LIF) (Merck Millipore, Cat#: ESG1107), supplemented with 2 μg/mL of doxycycline (dox) (Sigma-Aldrich, Cat#: 33429-100MG-R). iPSC medium supplemented with dox was replaced every alternate day after the first 3 days of reprogramming and withdrawn 4 days after the presence of iPSC-like colonies had formed, with typical dome-shaped iPSC morphology. Cells were then cultured to confluency on a layer of irradiated MEFs prior to further FACs purification and enrichment for Oct-GFP$^+$ cells. Purified *Oct4*-GFP iPSCs were then bulk expanded

in 175 cm² cell culture flasks (Corning, Cat#: CLS430825) and then frozen at a density of $1 \times 10^6$ cells/vial.

## Mouse embryonic fibroblast isolation and reprogramming

Reprogrammable mouse embryonic fibroblast (MEF) cultures were derived as described previously[102] from a E13.5dpc *Oct4*GFP-OKSM-M2rtTA embryo (animal I222e2) and cultivated at 37 °C, 5% O₂, 5% CO₂ in MEF medium containing DMEM High Glucose (Thermo Fisher Scientific, Cat# 11960-044) with 10% FBS (Thermo Fisher Scientific, Cat#: SH30071.03), 1 mM Sodium Pyruvate (Thermo Fisher Scientific, Cat#: 11360-070), GlutaMAX Supplement (Thermo Fisher Scientific, Cat#: 35050061), Penicillin-Streptomycin (Thermo Fisher Scientific, Cat#: 15070063), MEM Non-Essential Amino Acids Solution (Thermo Fisher Scientific, Cat#: 11140050) and 2-Mercaptoethanol (Thermo Fisher Scientific, Cat#: 21985023). MEFs were reprogrammed by being placed in iPSC medium supplemented with 2 μg/mL dox (Sigma-Aldrich, Cat#: 33429-100MG-R) and cultured on irradiated MEFs at 37 °C, 5% CO₂. iPSC colonies were discerned according to GFP expression in the absence of dox. In addition to bulk iPSC cultures (see below), single *Oct4*-GFP⁺ cells were deposited via FACS individually into 96-well pre-gelatinized tissue culture plates (Falcon, Cat#: 353072). Eighteen single-cell clones were bulk expanded on 6-well pre-gelatinized tissue culture plates (Falcon, Cat#: 353046) and maintained in serum or 2i conditions (see below).

### *Oct4*-GFP⁺ iPSC flow cytometry

For flow cytometry, cells were harvested by dissociating in 0.25% Trypsin EDTA (Life Technologies) to yield a single cell suspension, and then resuspended in FACS wash (Phosphate Buffered Saline with 2% Fetal Calf Serum) containing PI. Live cells were gated on the basis of forward scatter, side scatter and PI exclusion. Flow cytometric gates were set using control iPSCs that did not have endogenous GFP expression. Tubes were sorted according to GFP expression using an Influx Cell Sorter Instrument (Becton Dickinson). Data collected were analyzed and presented using FlowJo software version 10.8.1. Sorted GFP⁺ cells were then plated down on T-25 flasks (Corning) and expanded onto T-150 flasks (Corning), before being frozen down at a density of $1 \times 10^6$ cells/vial.

### Serum/LIF and 2i/LIF serum-free iPSC culture

Mouse iPSCs were maintained on irradiated primary MEFs, as previously described[8,103]. Briefly, iPSCs were cultured on 0.2% Porcine Gelatin (Sigma-Aldrich, Cat#: G1890-500G) coated tissue culture plates and flasks (Corning) on a feeder layer of irradiated MEFs ($2 \times 10^4$ cells/cm²). iPSC medium was changed daily and cells were cultured at 37 °C and 5% CO₂. Passaging was performed when iPSCs reached 70% confluency. Alternatively, iPSCs were cultured on irradiated MEFs in serum-free media containing knockout serum replacement (KOSR) and 2i/LIF[104]. Here, cells were cultured in DMEM (Thermo Fisher Scientific, Cat#: 11960-044), 1000 U/mL LIF (Merck Millipore, Cat#: ESG1107), 0.1 mM 2-Mercaptoethanol (Thermo Fisher Scientific, Cat#: 21985023), 1 mM GlutaMAX Supplement (Thermo Fisher Scientific, Cat#: 35050061), 1% Sodium Pyruvate (Thermo Fisher Scientific, Cat#: 11360-070), 0.1 mM MEM Non-Essential Amino Acids Solution (Thermo Fisher Scientific, Cat#: 11140050), 1% Penicillin-Streptomycin (Thermo Fisher Scientific, Cat#: 15070063), with medium supplemented with 15% KOSR (Thermo Fisher Scientific, Cat#: 10828-028), 1 μm Mek1/2 Inhibitor (PD0325901) (Tocris, Cat#: 4192) and 3 μm GSK3a/b inhibitor (CHIR99021) (Tocris, Cat#: 4423). Prior to genomic DNA extraction, iPSCs depleted from irradiated feeders were dissociated with 0.5% Trypsin EDTA (Thermo Fisher Scientific, Cat#: 25200-072). The irradiated MEFs were feeder depleted with 10 mL of iPSC media for 45 min in non-gelatinized T-25 flasks (Corning, Cat#: CLS3056). The resultant iPSCs were collected as a supernatant in a suspension medium.

## Lamivudine titration experiments

iPSCs were cultured with primary irradiated MEFs, as above, for 9 days in concentrations of lamivudine (3TC, Sigma-Aldrich, Cat#: L1295-10MG) ranging from 0 to 200 μM and cell survival calculated as a % of the starting population. Reprogrammable OKSM, rtTA3 MEFs were isolated from embryonic day 13.5 embryos from *Oct4*-GFP;ROSA-rtTA-out;OKSM-72 mice as previously described[44,105]. Doxycycline inducible reprogrammable MEFs were grown in media containing 2 μg/mL dox (Sigma Aldrich Cat#: 33429-100MG-R) and 0-200 μM 3TC for 15 days, with the percentage cell survival calculated at days 3, 7, 10 and 15. Once 100 μM was identified as the optimal concentration of 3TC to assess its impact on L1 retrotransposition, 30,000 reprogrammable MEFs at passage 2 were seeded onto gelatinized 6-well plates and reprogrammed in dox for 12 days, then cultured for an additional 4 days without dox. *Oct4*-GFP⁺ iPSCs were then purified via flow cytometric sorting and expanded on irradiated MEFs for an additional 11 days, then feeder depleted prior to DNA extraction for ONT sequencing. Reprogramming and iPSC media contained serum, and either 100 μM 3TC or no 3TC.

## Illumina sequencing and genomic analysis

Genomic DNA was harvested from MEFs and iPSCs using a DNeasy Blood and Tissue Kit (Qiagen, Cat#: 60594). DNA was quantified by a Qubit dsDNA HS Assay Kit (Life Technologies, Cat#: Q32851) on a Qubit Fluorometer 3.0 (Life Technologies). For WGS, libraries were generated using an Illumina TruSeq DNA PCR-free kit (Illumina, Cat#: 20015962) and sequenced separately on an Illumina HiSeq X Ten platform (Macrogen, Korea).

For mRC-seq, libraries were prepared as follows: 1 μg genomic DNA was sheared using a Covaris M220 Focused Ultrasonicator in a 130 μL microTUBE AFA fiber snap-cap vial (Covaris, Cat#: 520045). The following parameters were used to gain 500 bp insert libraries: 50 W, duty factor 20%, 200 cycles per burst, duration 55 s. Size selection to remove fragments <300 bp was performed using Agencourt AMPure XP beads (Beckman Coulter, Cat#: A63881) with a 1:0.6 DNA:beads ratio. Libraries were then generated by TruSeq Nano DNA LT kit (Illumina, Cat#: 20015964) using TruSeq DNA Single Indexes (Illumina, Cat#: 20015960 and 20015961) and run on a 2% agarose gel (Bioline, Cat#: BIO-41025) pre-stained with SYBR Safe Nucleic Acid Gel Stain (Invitrogen, Cat#: S33102). For ~500 bp insert size libraries the target gel fragment size was 600–650 bp, which was excised under a Safe Imager 2.0 Blue-Light Transilluminator (Invitrogen). DNA was purified using a MinElute Gel Extraction Kit (Qiagen, Cat#: 28606) according to the manufacturer's instructions. DNA was eluted in 25 μL molecular grade water. Enrichment of DNA fragments was performed as described for Illumina TruSeq Nano DNA LT Kit (Illumina, Cat#: 20015964). Sample clean up was performed with Agencourt AMPure XP beads (Beckman Coulter, Cat#: A63881) using a 1:1.1 ratio of DNA to beads. Amplified libraries were eluted in 30 μL molecular grade water and quantified using a Bioanalyzer DNA 1000 chip (Agilent Technologies, Cat#: 5067-1504).

mRC-seq hybridization was performed as previously described[28]. Hybridization reactions were washed using SeqCap Hybridization and Wash Kit (Roche, Cat#: 05634261001) and DNA eluted in 50 μL molecular grade water. Two post-hybridization LM-PCR reactions per sample were performed using 20 μL Enhanced PCR Mix, 5 μL PCR Primer Cocktail from the Illumina TruSeq Nano DNA LT Kit (Illumina, Cat#: 20015964) and 25 μL sample. PCR was performed with the following cycling conditions: 95 °C for 3 min, 8 cycles of 98 °C for 20 s, 60 °C for 15 s, and 72 °C for 30 s, followed by 72 °C for 5 min. The two PCR reactions for each sample were pooled and cleaned up using the QIAquick PCR Purification Kit (Qiagen) and samples eluted in 15 μL Elution Buffer (Qiagen, Cat#: 28706). Quantity and fragment size were determined using a Bioanalyzer DNA 1000 chip (Agilent Technologies, Cat#: 5067-1504). Libraries were

pooled and sequenced on an Illumina HiSeq X Ten platform (Macrogen, Korea).

Reads were aligned to the mm10 reference genome using bwamem[106] version 0.7.12 with parameters -M -Y. Duplicate reads were marked via Picard MarkDuplicates version 1.128. Indel Realignment was carried out via GATK IndelRealigner (3.7). SNVs were called by GATK HaplotypeCaller 3.7[54] to generate GVCFs and GenotypeGVCFs to obtain cohort-level calls. SNVs were also called using freebayes[55] filtered to remove known mouse strain germline variants[56]. SVs were called using Delly2 (version 0.7.9) and GRIDSS 2.0.0[57,58], using calls with concordant non-filtered precise breakends. SNVs and SVs, including potential private variants, were genotyped using the full catalog of variants obtained from all samples, including MEFs. Variant impact prediction and annotation was carried out using SnpEff version 4.3T[107]. WGS and mRC-seq aligned BAMs were processed to identify non-reference TE insertions using TEBreak[60] (https://github.com/adamewing/tebreak) version 1.1, as previously described[61].

## TE insertion PCR validation experiments
Reads supporting putative de novo TE insertions were manually examined using Serial Cloner (http://serialbasics.free.fr/Serial_Cloner.html) version 2.6, the UCSC Genome Browser BLAT tool[108] (version 438) and the Repbase CENSOR tool[109] (version 4.2.22). PCR primers were designed with Primer3[110] against TE insertion sequences and their 5′ and 3′ genomic flanks (Supplementary Data 3). Empty/filled PCRs (combining 5′ and 3′ flanking primers) and full-length PCRs (using junction-spanning primers) were performed using an Expand Long Range dNTPack (Roche, Cat#: 4829034001). Reaction mixes contained 5 µL 5× Expand Long Range Buffer with 12.5 mM MgCl₂, 1.25 µL dNTP Mix (dATP, dCTP, dGTP, dTTP at 10 mM each), 1.25 µL DMSO (100%), 1 µL primer mix (25 µM of each primer), 0.35 µL Expand Long Range Enzyme Mix (5 U/µL), 4–10 ng genomic DNA template, and molecular grade water up to a total volume of 25 µL. PCR was performed with the following cycling conditions: 92 °C for 3 min, 10 cycles of 92 °C for 30 s, 56–60 °C for 30 s, and 68 °C for 7 min 30 s 25 cycles of 92 °C for 30 s, 56–60 °C for 30 s, and 68 °C for 7 min + 20 s cycle elongation for each successive cycle, followed by 68 °C for 10 min. TE-genome junction validation PCRs were performed using MyTaq HS DNA Polymerase (Bioline, Cat#: BIO-2111). Reaction mixes contained 5 µL 5× MyTaq Reaction Buffer, 0.5 µL primer mix (25 µM of each primer), 0.2 µL MyTaq HS DNA Polymerase, 2–4 ng genomic DNA template, and molecular grade water up to a total volume of 25 µL. PCRs were performed using the following conditions: 95 °C for 2 min, 35 cycles of 95 °C for 15 s, 55/57 °C for 15 s, and 72 °C for 10 s, followed by 72 °C for 10 min. PCR products were run on 0.8–2% agarose gels (Bioline, Cat#: BIO-41025), depending on fragment size, pre-stained with SYBR Safe Nucleic Acid Gel Stain (Invitrogen, Cat#: S33102). A Typhoon FLA 9000 (GE Healthcare Life Sciences) was used for gel imaging. Gel fragments were excised under a Safe Imager 2.0 Blue-Light Transilluminator (Invitrogen). DNA purification was performed using the QIAquick Gel Extraction Kit (Qiagen, Cat#: 28706) or MinElute Gel Extraction Kit (Qiagen, Cat#: 28606) according to the manufacturer's instructions. PCR fragments were either sequenced directly or cloned using the pGEM-T Easy Vector System (Promega, Cat#: A1360) and Sanger sequenced to resolve insertion characteristics, as shown in Supplementary Data 3.

## L1-mCherry retrotransposition assays
The L1-mCherry construct was derived from the construct pTN201, a pCEP4-based vector containing the native mouse element L1_spa[22]. The L1_spa coding sequence was modified by site-directed mutagenesis to include two nonsynonymous nucleotide substitutions, rendering the ORF1p amino acid sequence identical to that of the L1 T_F subfamily consensus sequence[111,112]. The 3′UTR was interrupted by a reporter cassette based on previously described L1 retrotransposition indicator plasmids[19,113]. This reporter cassette consisted of the mCherry coding sequence in antisense orientation to the L1 and was equipped with an EF1α promoter and HSVtk polyadenylation signal. The mCherry ORF was interrupted by a β-globin intron oriented in a sense to the L1. The mCherry cassette was cloned using G-block double-stranded DNA fragments synthesized by Integrated DNA Technologies (IDT) and PCR products generated using Q5 DNA polymerase (New England Biolabs, Cat#: M0492). The mCherry coding sequence was synthesized with silent mutations ablating potential splice donor and splice acceptor sites that could interfere with intended splicing of the intron. In the L1-mCherry construct, the final 157 bp of the L1_spa 3′UTR, which included a conserved poly-purine tract, was situated downstream of the mCherry cassette and immediately upstream of the pCEP4 SV40 polyadenylation signal[112]. The L1-mCherry_RT- mutant contained a missense mutation in the reverse transcriptase domain of ORF2 (D709Y)[22]. Plasmids were prepared using a Qiagen Plasmid Plus Midi Kit and a QIAvac vacuum manifold (Qiagen, Cat#: 12145).

HeLa-JVM cells[19] were obtained from the laboratory of John V. Moran. These were cultured at 37 °C and 5% CO₂ in HeLa complete medium (DMEM, Life Technologies, Cat#: 11960044) supplemented with 10% FBS (Life Technologies, Cat#: 10099141), 1% Glutamax (Life Technologies, Cat#: 35050061) and 1% penicillin-streptomycin (Life Technologies, Cat#: 15140122). Cells were passaged at 70–80% confluency using 0.25% Trypsin-EDTA (Life Technologies, Cat#: 25200072). Cultured cell retrotransposition assays were then performed as described previously[67,113], except retrotransposition was detected by mCherry fluorescence instead of EGFP fluorescence. Briefly, $1 \times 10^5$ HeLa-JVM cells were seeded per well of a 6-well plate. Eighteen hours later, cells were transfected with 1 µg L1-mCherry or L1-mCherry_RT- plasmid per well using 3 µL FuGENE HD transfection reagent (Promega, Cat#: E2311) and 97 µL Opti-MEM (Life Technologies, Cat#: 31985047) per well according to the manufacturer's protocol. Twenty-four hours post-transfection, medium was replaced with either HeLa complete medium with 200 µg/mL Hygromycin (Life Technologies, Cat#: 10687010), or HeLa complete medium with 200 µg/mL Hygromycin and 100 µM Lamivudine (Sigma-Aldrich, Cat#: L1295-10MG). Medium was replaced every other day, and at 8 days post-transfection cells were collected by trypsinization, resuspended in sterile PBS, and analyzed on a CytoFLEX flow cytometer (Beckman Coulter) using the accompanying CytExpert software (version 2.5) to determine the percentage of mCherry positive cells. Three biological replicate assays were performed, each consisting of 3 assayed wells per condition (technical replicates). Mouse L1 retrotransposition reporter plasmids can be obtained from the corresponding authors.

## L1-mneoI retrotransposition assays
To prepare reporter constructs, miPSC_1_L1 and miPSC_4_L1 were amplified from genomic DNA using an Expand Long Range dNTPack (Roche, Cat#: 4829034001). Reaction mixes contained 5 µL 5× Expand Long Range Buffer with 12.5 mM MgCl₂, 1.25 µL dNTP Mix (dATP, dCTP, dGTP, dTTP at 10 mM each), 1.25 µL DMSO (100%), 1 µL primer mix (50 µM of each primer), 0.35 µL Expand Long Range Enzyme Mix (5U/µL), 10 ng genomic DNA template and molecular grade water, up to a total volume of 25 µL. PCRs were performed with the following cycling conditions: 92 °C for 3 min, 10 cycles of 92 °C for 30 s, 58 °C for 30 s, and 68 °C for 7 min 30 s; 25 cycles of 92 °C for 30 sec, 58 °C for 30 s, and 68 °C for 7 min plus 20 s elongation for each successive cycle, followed by 68 °C for 10 min. Primers introduced a NotI restriction site at the L1 5′ end (miPSC_1_L1_F, 5′-tttgcggccgcagaaagggaataatcgaggtg-3′; miPSC_1_L1_R, 5′-gctaagcttgag aataagtgaagga-3′; miPSC_4_L1_F, 5′-agggcggccgcaggattaagaacccaa tcaccag-3′; miPSC_4_L1_R, 5′-aaaatgcctgttgtgtgccaat-3′). Reactions were purified using agarose gel electrophoresis. Target fragments were excised and purified using either traditional phenol-chloroform extraction or QIAquick and MinElute Gel Extraction Kits (Qiagen,

Cat#: 28706 and 28604). Each L1 was then cloned into pGEMT Easy Vector (Promega, Cat#: A1360). Ligations were incubated overnight at 4 °C. Ligation reactions were transformed using One Shot TOP10 chemically competent *E. coli* (Invitrogen, Cat#: C404010). Blue/white screening was performed using LB/ampicillin/IPTG/X-Gal plates. At least 3 positive colonies per L1 were chosen for Miniprep culture and plasmid DNA was isolated using a QIAprep Spin Miniprep Kit (Qiagen, Cat#: 27106). At least three clones per element were capillary sequenced and compared to identify PCR-induced mutations. Full-length L1s were then reconstructed by combining PCR-mutation free fragments from different clones using restriction enzymes (New England Biolabs) recognizing the L1 sequence. Reactions were purified using agarose gel electrophoresis and target fragments were excised and purified using QIAquick and MinElute Gel Extraction Kits (Qiagen, Cat#: 28706 and 28604).

pTN201 was used to generate L1 reporter constructs. pTN201 is composed of a pCEP4 backbone (Life Technologies) containing L1$_{spa}$, a retrotransposition-competent L1 T$_F$[22] and a downstream mneoI retrotransposition reporter cassette[114]. The mneoI cassette is driven by an SV40 promoter and holds the neomycin resistance gene, which is interrupted by an intron and is positioned antisense to L1spa. In this assay, neomycin (or its analog, Geneticin/G418) resistance only occurs via transcription, splicing and integration of the L1 and mneoI cassette into genomic DNA[19,67]. To measure miPSC_1_L1 and miPSC_4_L1 retrotransposition efficiency, L1$_{spa}$ was removed from the pCEP4 backbone by digesting with *NotI* and *PacI*. The pCEP4 backbone was dephosphorylated using Calf Intestinal Alkaline Phosphatase (CIP) (New England Biolabs, Cat#: M0290). The backbone and fragments of either miPSC_1_L1 or miPSC_4_L1 were combined in a single ligation reaction using T4 DNA Ligase (New England Biolabs, Cat#: M0202) and incubated overnight at 16 °C. Ligations were transformed using One Shot TOP10 chemically competent *E. coli* (Invitrogen, Cat#: C404010) and plasmid DNA of positive clones was obtained using QIAprep Spin Miniprep Kit (Qiagen, Cat#: 27106). Clones were verified as mutation-free by capillary sequencing. Plasmid DNA for retrotransposition assays was obtained using a Plasmid Maxi Kit (Qiagen, Cat#: 12163). Each construct was built with and without a cytomegalovirus promoter (CMVp) preceding the L1. In addition, the following controls, each based on a pCEP4 backbone containing the mneoI cassette, were employed: TGF21, a retrotransposition-competent L1 G$_F$;[21] L1SM, a synthetic codon optimized mouse L1;[68] L1SMmut2, L1SM immobilized by reverse transcriptase and endonuclease domains mutations[68].

Retrotransposition assays were performed as previously described[67], with minor modifications. HeLa-JVM cells were grown in HeLa complete medium (DMEM, Life Technologies, Cat#: 11960044) supplemented with 10% FBS (Life Technologies, Cat#: 10099141), 1% Glutamax (Life Technologies, Cat#: 35050061) and 1% penicillin-streptomycin (Life Technologies, Cat#: 15140122), and then seeded at a density of $4 \times 10^4$ cells/well in 6-well tissue culture plates. 14–16 h after plating, cells were transfected with L1 reporter constructs using 4 µL FuGENE HD transfection reagent (Promega, Cat#: E2311) 96 µL Opti-MEM (Life Technologies, Cat#: 31985047) and 1 µg plasmid DNA per well. Transfection efficiencies were determined in parallel by preparing transfection mixes containing 4 µL FuGENE HD transfection reagent (Promega, Cat#: E2311), 96 µL Opti-MEM (Life Technologies, Cat#: 31985047), 0.5 µg L1 expression plasmid and 0.5 µg pCEP4-eGFP. The transfection mixture was added to each well containing 2 mL DMEM-complete medium. Plates were incubated at 37 °C and 5% CO$_2$, medium replaced 24 h post-transfection, and transfection efficiency determined 72 h post-transfection. pCEP4-eGFP transfected wells were trypsinized and cells were collected from each well and centrifuged at 2000g for 5 min. Cell pellets were resuspended in 300–500 µL 1× PBS. The number of eGFP-positive cells was determined using a CytoFLEX flow cytometer (Beckman Coulter). The percentage of eGFP-positive cells was used to normalize the G418-resistant colony counts for each

L1 reporter construct[67]. G418 (400 µg/mL) (Thermo Fisher Scientific, Cat#: 10131035) selection was started 3 days post-transfection and performed for 12 days. G418 foci were washed with 1× PBS and fixed using 2% Formaldehyde/0.2% Glutaraldehyde in 1× PBS (Sigma-Aldrich) fixing solution at room temperature for 30 min. Staining was done using 0.1% Crystal Violet solution (Sigma-Aldrich) at room temperature for 10 min. Foci were counted in each well to quantify retrotransposition.

## L1 bisulfite sequencing experiments

Bisulfite conversion was performed with 200 ng input genomic DNA from miPSC lines and MEFs using a EZ DNA Methylation-Lightning Kit (Zymo Research, Cat#: D5030), following the manufacturer's instructions. DNA was eluted in 10 µL Elution Buffer. The internal sequences of L1 T$_F$ monomers were amplified genome-wide with the following primers: BS_TflII_mono_F, 5′-GGAAATTAGTTTGAATAGGTTAGAGGGTG; BS_TflII_mono_R, 5′-TCCTAAATTCCAAAAAATCCTAAAACCAAA. The following locus-specific primers were used to target the 5′ promoter region of the following elements of interest: BS_miPSC_1_L1_F, 5′-TGAT TTATTTTTGATTGAATTTATTTTTAT; BS_miPSC_1_L1_R/donor_L1_R, 5′-CTATTCAAACTAATTTCCTAAATTCTACTA; BS_miPSC_3_L1_F, 5′-TAGT TGGGGGTTGTATGATGTAAGTT; BS_miPSC_3_L1_R, 5′-TCCCAAAAACTA TCTAATTCTCTAAC; BS_miPSC_4_L1_F, 5′-TTTATATTGAAGGTTTGGAT GATTTTATAT; BS_miPSC_4_L1_R, 5′-TCCAATTCTCTAATACACCCTCT AAC; BS_donor_L1_F, 5′-TTAAAGAAGTTAGTGATTTTTTAGAATTTT.

PCRs were performed using MyTaq HS DNA Polymerase (Bioline, Cat#: BIO-21111). Reaction mixes contained 5 µL 5× MyTaq Reaction Buffer, 0.5 µL primer mix (25 µM of each primer), 0.2 µL MyTaq HS DNA Polymerase, DMSO at a final concentration of 0.1%, 2 µL bisulfite converted DNA template, and molecular grade water up to a total volume of 25 µL. PCR cycling parameters were as follows: 95 °C for 2 min, 40 cycles of 95 °C for 30 s, 54 °C for 30 s, and 72 °C for 30 s, followed by 72 °C for 5 min. PCR products were run on a 2% agarose gel, excised and purified using a MinElute Gel Extraction Kit (Qiagen, Cat#: 28604) according to the manufacturer's instructions. Illumina libraries were constructed using a NEBNext Ultra™ II DNA Library Prep Kit (New England Biolabs, Cat#: E7645). Libraries were quantified using a Bioanalyzer DNA 1000 chip (Agilent Technologies, Cat#: 5067-1504). Barcoded libraries were pooled in equimolar amounts and sequenced as 2x300mer reads on an Illumina MiSeq platform using a MiSeq Reagent Kit v3 (Illumina, Cat#: MS-102-3003). 50% PhiX Control v3 (Illumina, Cat#: FC-110-3001) was used as a spike-in. Sequencing data were analyzed as described previously[34]. To summarize, for the L1 T$_F$ genome-wide analysis, paired-end reads were considered separately and those with the L1 T$_F$ bisulfite PCR primers at their termini were retained and aligned to the mock converted T$_F$ monomer target amplicon sequence with blastn. Reads where non-CpG cytosine bisulfite conversion was <95%, or ≥5% of CpG dinucleotides were mutated, or ≥5% of adenine and guanine nucleotides were mutated, were removed. 50 reads per sample, excluding identical bisulfite sequences, were randomly selected and analyzed using QUMA[115] version 1.1.16 with default parameters, with strict CpG recognition. Specific L1 loci were analyzed in a similar fashion, except paired-end reads were assembled into contigs, as described elsewhere[34], prior to blastn alignment to the mock converted L1 locus target amplicon.

## Nanopore sequencing analyses

Genomic DNA was extracted from 6 miPSC lines reprogrammed in the presence of 100 µM 3TC, 6 miPSC lines generated without 3TC, and the parental MEFs, with a Nanobind CBB Big DNA Kit (Circulomics, Cat#: NB-900-001-01) according to the manufacturer's instructions. DNA libraries were prepared at the Kinghorn Centre for Clinical Genomics (KCCG, Australia) using 3 µg input DNA, without shearing, and an SQK-LSK110 ligation sequencing kit. Libraries were each sequenced separately on a PromethION (Oxford Nanopore Technologies) flow cell

(FLO-PRO002, R9.4.1 chemistry) (Supplementary Data 1). Bases were called with guppy 5.0.13 (Oxford Nanopore Technologies).

Non-reference TE insertions were detected with TLDR[35]. Briefly, this involved aligning ONT reads to the mm10 reference genome using minimap2[116] version 2.20 (index parameter: -x map-ont; alignment parameters: -ax map-ont -L -t 32) and SAMtools[117] version 1.12. BAM files were then processed as a group with TLDR[35] version 1.2.2 (parameters -e teref.mouse.fa -p 128 -m 1 -r mm10.fa -n nonref.collection.mm10.chr.bed.gz --keep_pickles). The files teref.mouse.fa, composed of TE family consensus sequences, and nonref.collection.mm10.chr.bed.gz, a collection of known nonreference retrotransposon insertions, are available from github.com/adamewing/tldr/. The TLDR output table was further processed to remove calls not passing relevant TLDR filters, where family = "NA" or remappable = "FALSE" or UnmapCover <0.5 or LengthIns <100 or EndTE-StartTE <100 or strand = "None" or SpanReads <1. 3′ truncated TE insertions, and B1 or B2 insertions 5′ truncated by more than 2 bp, were removed. Events detected in only one miPSC line and not matching a known non-reference insertion were designated as putative de novo insertions (Supplementary Data 3).

Reference TE methylation was assessed for parental MEFs and miPSC lines aggregated by condition (reprogrammed with or without 3TC), using Methylartist version 1.2.4[82]. Briefly, CpG methylation calls were generated from ONT reads using nanopolish version 0.13.2[118]. Using Methylartist commands db-nanopolish, segmeth and segplot with default parameters, methylation statistics were generated for the genome divided into 10kbp bins, protein-coding gene promoters defined the Eukaryotic Promoter Database (−1000bp,+500 bp)[119], and reference TEs defined by RepeatMasker coordinates. TE families displayed in Fig. 4a included $T_F$, $G_F$, and A-type L1s > 6kbp, B1 and B2 SINEs, and MERVL MT2 and IAP elements represented by their long terminal repeats. Methylation values were calculated for L1 5′UTRs only, excluding the L1 body. Methylation profiles for individual loci were generated using the Methylartist command locus. L1 $T_F$ methylation profiles shown in Fig. 4b were generated for elements >7kbp with the Methylartist command composite. To identify individual differentially methylated TEs (Supplementary Data 5), we required elements to have at least 4 reads and 20 methylation calls in each of the MEF, aggregated control miPSC and aggregated 3TC-treated miPSC datasets. Statistical comparisons were performed based on methylated and unmethylated CpG call counts, using Fisher's exact test with Bonferroni correction for multiple testing.

L1-mCherry retrotransposition events were recovered via ONT sequencing in a similar fashion to the analysis of mouse samples, except genomic DNA obtained from 5 HeLa cell lines expanded over 3–5 passages from single L1-mCherry insertion-harboring colonies was barcoded using a SQK-MLK111.96-XL kit, pooled in equimolar amounts, and sequenced on a single PromethION flow cell. Reads were aligned to the hg38 human reference genome build. L1-mCherry insertions were then identified using TLDR, as for the endogenous mouse TE analysis described above.

**Statistics and reproducibility**
Statistical analyses were performed using GraphPad Prism version 9, with the exception of Supplementary Fig. 1 (Seaborn version 0.9) and Supplementary Data 5 (SciPy version 1.4.1). Data were presented in histograms as mean ± SD and replicate (n) values provided in the corresponding figure legends. No statistical method was used to predetermine sample size. Experiments were designed to obtain at least biological triplicate data. No data were excluded from the analysis. One miPSC line, from animal A172 astrocytes, generated insufficient cells for subsequent genomic analyses. Experiments were not randomized and the investigators were not blinded to sample identity. PCR experiments to validate TE insertions were repeated independently at least twice, with on-target amplicons (as shown in Fig. 1b–e, Fig. 2b–f,

Fig. 3e, Supplementary Fig. 3, and Supplementary Fig. 9) confirmed by capillary sequencing. Retrotransposition assays, as shown in Fig. 1h and Fig. 3b, were performed on three different days (independent biological replicates) with three wells per assay (technical replicates).

**Reporting summary**
Further information on research design is available in the Nature Portfolio Reporting Summary linked to this article.

## Data availability
All Oxford Nanopore Technologies and Illumina sequencing data generated by this study are available from the European Nucleotide Archive (ENA) under project accession code "PRJEB20569". Gene promoter coordinates were obtained from the Eukaryotic Promoter Database (https://epd.epfl.ch/) and reference TE coordinates were defined by the UCSC Genome Browser RepeatMasker track (https://hgdownload.soe.ucsc.edu/goldenPath/mm10/). All other relevant data supporting the key findings of this study are available within the article and its Supplementary Information files or from the corresponding author upon reasonable request. Source data are provided with this paper.

## Code availability
TEBreak, TLDR and Methylartist, and instructions for their use, are available at https://github.com/adamewing/tebreak, https://github.com/adamewing/TLDR and https://github.com/adamewing/methylartist, respectively.

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

## Acknowledgements

The authors thank Jef D. Boeke and John V. Moran for sharing L1SM plasmids and the HeLa-JVM cell line, respectively. This study was supported by an Australian Government Research Training Program (RTP) Scholarship and a Mater Research Frank Clair Scholarship awarded to P.G., ARC Discovery Early Career Researcher Award (DE150101117), Discovery Project (DP170101198) and Australian Department of Health Medical Frontiers Future Fund (MRFF) (MRF1175457) grants awarded to A.D.E., NHMRC Project Grant (GNT1051117), ARC Future Fellowship (FT180100674) and Sylvia and Charles Viertel Senior Medical Research Fellowship funds awarded to J.M.P., NHMRC Investigator Grants (GNT1178460 to R.L., GNT1173476 to S.R.R., GNT1173711 to G.J.F.), and CSL Centenary Fellowship and NHMRC Project Grant (GNT1106206, GNT1125645, GNT1126393, GNT1138795) funding awarded to G.J.F. A.D.E., S.R.R. and G.J.F. additionally acknowledge support from the Mater Foundation.

## Author contributions

P.G., S.M.L., M.R.L., D.C., F.J.S-L., L.W., A.L.C., C.J., A.S.K., P.E.C., C.M.N. and S.R.R. performed experiments. R.L., J.M.P. and G.J.F. provided resources. A.D.E. and G.J.F. performed bioinformatic analyses. P.G., S.M.L., S.R.R. and G.J.F. generated figures. P.G., S.R.R., and G.J.F. conceived the study. P.G., S.M.L., S.R.R., J.M.P, and G.J.F. designed and supervised experiments. G.J.F. wrote the manuscript and directed the study. All authors commented on the manuscript.

## Competing interests

The authors declare no competing interests.
