## [Peer Review File · Nature Communications]

REVIEWER COMMENTS

Reviewer #1 (Remarks to the Author):

In the study by Gerdes et al., iPS cells were generated with transgenic mice that can conditionally express the Yamanaka factors to reprogram cells towards iPS cells. They have used this system to investigate the effect of the reprogramming process on genetic variation and then in particular, that caused by retrotransposition events, although some attention is also paid to SVs and SBS. The authors have used both bulk approaches and examined clonal lines and used different sequencing technologies including WGS, mRC-seq (which uses capture probes) and oxford nanopore technology (ONT) to identify de novo TE insertions. This study is of particular relevance if translatable to human iPSCs, because such events may lead to undesired phenotypes and impede their clinical use. However, it is difficult to understand how the observations translate to human cells, because of the species differences in TE numbers, but possibly also because of differences in the biology of regulation of TE mobilisation and the differences between mouse versus human pluripotent stem cells. Moreover, the experimental set-up using bulk populations to start with is not ideal for the research question at hand. The manuscript would also improve greatly if the effect of 3TC on retrotransposition events during reprogramming was examined in more detail (and not just by ONT) to demonstrate the feasibility of this approach in minimising TE events as a result of reprogramming.

Major comments:

The authors mention that the early embryo is a niche for retrotransposition events. This means that depending on the timing of these events and the eventual contribution of the cells in which these events occurred to development, there can be high or low degrees of somatic mosaicism of the retrotransposition events of embryonic origin. While I agree that the majority of the events detected by the authors are likely de novo, determine if the measured variant was caused prior to, during or after the reprogramming process. The study would therefore have been a lot stronger if a clonal step was performed prior to reprogramming. Although this issue has to a certain extent been addressed by a down-sampling approach, the study would improve if addressed experimentally.

In line with this, the authors should demonstrate at what VAF the mRC-seq is still able to detect retrotransposons. This is important for the conclusion that the event was not present in the parental population

Also in line with these comment: the comparison serum vs 2i (that cells have been exposed to for 3 passages) does not make sense to me if not followed by a clonal step, because any event that occurs within the 3 passages in a bulk population are by definition present at low frequency, so probably hard to detect. It is likely that all events that have been detected occurred prior to the change in growth conditions.

For the SBS and SVs, the authors used variants from a database as a reference whereas they have also generated sequencing data from the embryonic fibroblasts. These could have better been used as a reference (see also <https://doi.org/10.1038/nprot.2017.111>). Using external databases as germline reference is not ideal, because mouse strains will accumulate mutations that are not in the databases. Moreover, because samples were processed in bulk, most of the somatic mutations will be missed because of low VAF. The quantification of genetic variants (SNVs, SVs, etc) will be highly inaccurate. Ideally the authors would have performed a clonal step prior to reprogramming followed by another clonal step after reprogramming to accurately and highly sensitively identify all the variants induced by the reprogramming process itself (see also

<https://doi.org/10.1038/nprot.2017.111>). In my opinion, the authors can therefore not conclude that choice of primary cell type does not significantly impact the frequency of SNVs and SVs later found in miPSC lines.

In a similar fashion, no conclusion can be drawn on the effects of 2i or serum media on the accumulation of SNVs or SVs (lines 147-149), because this requires 2 consecutive clonal steps and a period in between to allow cells to accumulate mutations. This sentence and other comments on private SNVs/SVs can better be removed from the manuscript.

The RT inhibitor lamivudine does not inhibit programming. The study would be more interesting if shown that lamivudine leads to fewer retrotransposition events.

The authors also performed Nanopore sequencing of 4 bulk iPS cell lines and detected 16 TE insertions, one of which was also present in the feeders and one in the parental MEFs, leaving 14 de novo events. If I understand correctly, none of these events could be validated by PCR (0/14), while the success-rate for the de novo events that had been detected with other methods (mRC-seq + WGS) was much higher (38/41). Please explain this discrepancy and use a second method (PCR or otherwise) to validate the events detected by ONT.

In line with this, were flanking genomic sequences detected by ONT? In other words, are these real transposition events or could it be reverse transcribed RNA that has not integrated?

The authors mention several differences between mice and men. With those differences in mind, how do the observations translate to human cells?

Minor comments:

In the abstract it is said that mouse iPSCs are able to silence transposable elements. This statement seems to be in conflict with some other comments in the manuscript: e.g. in the final paragraph of the introduction: "Mouse ESCs cultured in standard media containing serum express endogenous L1 proteins and support engineered L1 mobilisation. Naïve ESCs grown in media containing two small-molecule kinase inhibitors (2i) in place of serum also exhibit L1 promoter hypomethylation." Possibly better to rephrase the abstract.

I do not see the added value of the bulk experiments over the clonal analyses and would remove those from the manuscript.

In figure 1g only iPS cell lines are shown. This figure would improve if data of the parental cells is also included (the somatic cells after isolation prior to reprogramming) as a control for the bisulfite conversion.

In figure 1h a retrotransposition assay is conducted. This data is rather confusing, because 2 different cell types are presented in the same graph (HeLa + iPS cells) that cannot be compared with one another (see also discussion on species differences). In these experiments, cells are transfected with a reporter for retrotransposition events. According to the legend, the data has been normalised to L1 (in HeLa cells). Shouldn't the data of the iPS cells be presented separately and normalised against L1 in iPS cells instead of the L1 in HeLa cells? Moreover, in the methods section, the experiments are being discussed for the HeLa cells, but not the iPSCs.

The rationale between the two different types of media (2i versus serum) is not entirely clear

Lines 152-154: Why were some events detected by mRC-seq and not by WGS and vice versa? Were the mRC-seq only events visible in the WGS data e.g. in IGV?

Reviewer #2 (Remarks to the Author):

In this manuscript, Gerdes et al., study the role of retrotransposons as main drivers of the mutational landscape in mouse-induced pluripotent stem cells (miPSCs). This is a very interesting and comprehensive study that combines state-of-the-art methods in order to re-evaluate the role of retrotransposon integration in miPSCs. It is of importance to the field of transposable elements and regenerative medicine.

The scientists reprogrammed mice that expressed a doxycycline-inducible reprogramming cassette. They used tissues of all 3 germ layers in order to produce miPSCs and looked for TE insertions by Illumina sequencing using WGS and mRC-seq. They also characterised the de novo inserted TE sequences. Additionally, the researchers used single-cell isolated clones for cell expansion to look at the TE de novo insertions in a more homogenous cell population. They found a polymorphic retrotransposition competent L1 that evades DNA methylation. Interestingly, inhibiting the L1 reverse transcription with lamivudine to prevent potential de novo mutations had no effect on reprogramming activity. Finally, Gerdes et al confirmed genomic TE insertions with nanopore sequencing in bulk miPSCs. Nanopore sequencing also allowed them to look at genome-wide DNA demethylation during iPSC reprogramming and the researchers found young L1s to lose more methylation than other sites.

Major comments:

- I am not sure how the scientists evaluated the reprogramming efficiency of the miPSCs that were treated with lamivudine or not. The scientists should also evaluate the general demethylation dynamics in the lamivudine treated samples (similar to Fig 3c) in order to see whether epigenetic reprogramming is at all affected.**
- In the nanopore sample Gerdes et al. still found 2 L1s that are able to evade RT inhibition with lamivudine. This needs to be added to the discussion and also rephrased in the abstract, because of course if inhibition of L1 RT is not sufficient the question is whether reprogramming is affected if you were to inhibit L1 RT to 100%. The researchers state that this result is not statistically significant because of the low sample number however the question is also the depth of the nanopore sequencing. Does this allow to catch all the L1s that are inserted de novo?**
- I would like the scientists to comment on the fact that they did not find ERV elements in the short-read data but found them in the long read data.**
- What is the depth of the nanopore sequencing and what is the exact overlap between nanopore sequencing and WGS or RC-Seq? Did you only take forward reads that overlap both WGS and had at least 1 read in nanopore? Do you think there are more insertions that you missed because of the depth of nanopore sequencing which could not be found in WGS and RC-seq?**
- Does the reprogramming efficiency differ for the different cell types? The scientist should report the numbers of original reprogramming efficiency from their FACS sort of GFP positive cells. Does this lead to a differing number of L1s?**

Reviewer #3 (Remarks to the Author):

General comments:

The authors focus on a possible drawback of the use of somatic cell reprogramming to derive mouse-induced PSCs which was never fully addressed before: the vulnerability of miPSCs to TE-mediated mutagenesis due to TE mobility during reprogramming. Mutagenesis

of miPSCs could impair their applications in many fields thus, it is important to understand and investigate all the possible sources of reprogramming-derived/mediated mutagenesis and the possibility to counteract. I liked that the authors addressed these points in their work thus I value positively this paper, but more effort in addressing the Lamivudine data would give more strength, in my opinion.

In general, I find the paper to be clear and linear: the results follow a logical flow, I think they are well described in the text and by the figures, captions are very well written (all details are present). Methods are exhaustive. Discussion highlights the future steps that can be taken.

However, I can't really comment too much on the novelty and the strength of the techniques used, I would add in the introduction a conclusive section that describes what the authors found and their contribution to the field.

Other comments:

1. In figure 1A I would change the name of the mouse model to make it clearer.
2. PCR controls in figure 1: would it be good to add as negative control the parental line and not MEFs only?
3. Figure 1C: there are some faint bands. Are they no-specific bands or what?
4. They say Lamivudine doesn't affect the reprogramming efficiency, however, to support this claim they provide only the data about cell survival (supplementary figure 7). After the reprogramming, I think the quantification of OCT4+ cells and additional markers would be preferable to support such a conclusion.
5. They didn't observe any significant difference between TE transpositions after reprogramming with or without Lamivudine, however, they state that there might be an effect of Lamivudine. I'm suggesting making this result more redundant and stronger. Either repeat the experiment (to increase the N) or try different strategies. It would be a very cool result and strengthen the entire work.
6. Would it be possible to understand the probability of L1 insertions in gene exons or gene regulatory elements based on the frequency of Degenerate L1 endonucleases motifs within the genome? Thus, to comment on the real probability that L1 transpositions can really generate functional mutations?

We thank the referees for their time and constructive feedback. On revision, we have added significant data, including Oxford Nanopore Technologies (ONT) long-read sequencing as well as reprogramming experiment quality control results, and have performed several new analyses. We divided the original Fig. 3, making a new Fig. 4, to make space for additional panels and added a supplemental figure, Extended Data Fig. 7. New data can therefore be found in Fig. 1a, Fig. 3c-g, Fig. 4a-c, Extended Data Fig. 7 and 9, and Supplementary Tables 1, 3, 4 and 5. Please see our point-by-point responses below.

Reviewer #1 (Remarks to the Author):

In the study by Gerdes et al., iPSC cells were generated with transgenic mice that can conditionally express the Yamanaka factors to reprogram cells towards iPSC cells. They have used this system to investigate the effect of the reprogramming process on genetic variation and then in particular, that caused by retrotransposition events, although some attention is also paid to SVs and SBS. The authors have used both bulk approaches and examined clonal lines and used different sequencing technologies including WGS, mRC-seq (which uses capture probes) and oxford nanopore technology (ONT) to identify de novo TE insertions. This study is of particular relevance if translatable to human iPSCs, because such events may lead to undesired phenotypes and impede their clinical use. However, it is difficult to understand how the observations translate to human cells, because of the species differences in TE numbers, but possibly also because of differences in the biology of regulation of TE mobilisation and the differences between mouse versus human pluripotent stem cells. Moreover, the experimental set-up using bulk populations to start with is not ideal for the research question at hand. The manuscript would also improve greatly if the effect of 3TC on retrotransposition events during reprogramming was examined in more detail (and not just by ONT) to demonstrate the feasibility of this approach in minimising TE events as a result of reprogramming.

Major comments:

The authors mention that the early embryo is a niche for retrotransposition events. This means that depending on the timing of these events and the eventual contribution of the cells in which these events occurred to development, there can be high or low degrees of somatic mosaicism of the retrotransposition events of embryonic origin. While I agree that the majority of the events detected by the authors are likely de novo, determine if the measured variant was caused prior to, during or after the reprogramming process. The study would therefore have been a lot stronger if a clonal step was performed prior to reprogramming. Although this issue has to a certain extent been addressed by a down-sampling approach, the study would improve if addressed experimentally.

We appreciate the detailed comments provided by Reviewer #1. One overarching point they make here and below is that our study would have been stronger if a mouse embryonic fibroblast (MEF) clonal expansion step from single cells had been performed prior to the generation of miPSCs. Clonal expansion of this type is feasible for the adult stem cell experiments conducted by the Cuppen lab, as noted by Reviewer #1 (PMID:32427826;

PMID: 29215633), and we agree this suggestion has conceptual merit, which is why we performed the down-sampling they mention (Results, line 184). This approach is however not applicable to the present study. Firstly, the reprogrammable MEFs used here cannot be clonally expanded from a single cell and can no longer be induced to a pluripotent state after more than 3 passages of cell culture (PMID: 25225958; PMID: 19668190). Secondly, findings from clonal MEF cultures would arguably be less relevant to the vast majority of miPSC lines established as research models, which are derived from bulk primary cell cultures. Thirdly, even if we had been able to expand MEFs from single cells prior to reprogramming, they would still need to be extensively cultured prior to reprogramming, allowing more time for mutations to accumulate during this window. For these reasons, we have not performed the MEF clonal expansion step as requested by Reviewer #1. We do however explain why this is the case in the Discussion, and cited the above mentioned study (PMID:32427826) by noting on line 332 that: “It is nonetheless possible that additional somatic variants would have been annotated if primary single-cell clones, as analyzed elsewhere⁸⁶, were reprogrammed. However, the introduction of multiple bottlenecks followed by clonal expansion prolongs cell culture and could thereby exaggerate mutation frequencies. The parental MEFs used here cannot in any case be clonally expanded from single cells, and reprogram extremely inefficiently after more than three passages in culture^{87,88}, and for these reasons we did not prepare single-cell MEF clones prior to reprogramming.”

In line with this, the authors should demonstrate at what VAF the mRC-seq is still able to detect retrotransposons. This is important for the conclusion that the event was not present in the parental population

The vast majority of detected insertions were found by both WGS and mRC-seq. However, as requested, we provide these sensitivity statistics for mRC-seq on line 184 of the revised manuscript by noting: “Down-sampling followed by seeking at least one WGS or mRC-seq read in support of these non-reference insertions suggested our approach would distinguish approximately 50%, 95% and 99% of *de novo* TE insertions from pre-existing subclonal TE insertions present in 1%, 5% and 10% of cells, respectively (**Extended Data Fig. 4b**), whereas mRC-seq alone would achieve respective sensitivities of approximately 22%, 69% and 89%.”

Also in line with these comment: the comparison serum vs 2i (that cells have been exposed to for 3 passages) does not make sense to me if not followed by a clonal step, because any event that occurs within the 3 passages in a bulk population are by definition present at low frequency, so probably hard to detect. It is likely that all events that have been detected occurred prior to the change in growth conditions.

That was our conclusion as well. The vast majority of retrotransposition events occurred prior to the change in growth conditions, likely during reprogramming. As suggested, an additional clonal expansion step here *may* have revealed a difference in the number of retrotransposition events occurring during cell culture in 2i conditions, based on Reviewer #1’s assumption that such events would otherwise be hard to detect. However, we did not claim a difference in retrotransposition frequency when 2i and serum conditions were compared, even if

cultivation in 2i did significantly reduce L1 promoter methylation (as per **Fig. 2**). The experiment Reviewer #1 requests here would require clonal expansion from single miPSCs in 2i conditions, which again is not possible for these cells, which is why they were expanded first in serum for three passages, and then in 2i for three additional passages.

For the SBS and SVs, the authors used variants from a database as a reference whereas they have also generated sequencing data from the embryonic fibroblasts. These could have better been used as a reference (see also <https://doi.org/10.1038/nprot.2017.111>). Using external databases as germline reference is not ideal, because mouse strains will accumulate mutations that are not in the databases. Moreover, because samples were processed in bulk, most of the somatic mutations will be missed because of low VAF. The quantification of genetic variants (SNVs, SVs, etc) will be highly inaccurate. Ideally the authors would have performed a clonal step prior to reprogramming followed by another clonal step after reprogramming to accurately and highly sensitively identify all the variants induced by the reprogramming process itself (see also <https://doi.org/10.1038/nprot.2017.111>). In my opinion, the authors can therefore not conclude that choice of primary cell type does not significantly impact the frequency of SNVs and SVs later found in miPSC lines.

There are two important points here. The first, the use of the MEF WGS datasets to filter somatic SVs and SNVs called in the bulk miPSC lines, is actually what we did (as shown in **Extended Data Fig. 1**), although we did not explicitly say this in the text. We apologise for this oversight and now clarify in the Methods and revised main text that this is how the analysis was done. The relevant part of the Results section (line 78) now reads: “**Concordant SNVs detected by GATK HaplotypeCaller and freebayes^{54,55} were filtered to remove known mouse strain germline variants⁵⁶, yielding 3,603 SNVs private to a single miPSC line (average ~140 per line) and absent from the corresponding MEF samples (Supplementary Table 2).**” The relevant part of the Methods section (line 593) now reads: “**SNVs and SVs, including potential private variants, were genotyped using the full catalog of variants obtained from all samples, including MEFs.**”

The second point, highlighting the value of a clonal step prior to reprogramming, again based on experiments done by the Cuppen lab (PMID:32427826; PMID: 29215633) with adult stem cells, is not applicable to the vast majority of primary cell types used in the present study, which cannot be efficiently expanded from single cells prior to reprogramming. Ultimately, our conclusion that the number of SNVs and SVs was not different amongst the bulk reprogrammed miPSC lines, or those obtained as single-cell miPSC clones from MEFs, is supported by the available data. Any other conclusion would be based on the assumed outcome of experiments that we did not, and could not, perform. We explain this further in the Discussion (line 332), cite one of the above studies, and note that: “**It is nonetheless possible that additional somatic variants would have been annotated if primary single-cell clones, as analyzed elsewhere⁸⁶, were reprogrammed. However, the introduction of multiple bottlenecks followed by clonal expansion prolongs cell culture and could thereby exaggerate mutation frequencies. The parental MEFs used here cannot in any case be clonally expanded from single cells, and reprogram extremely inefficiently after more than three passages in**

culture^{87,88}, and for these reasons we did not prepare single-cell MEF clones prior to reprogramming.”

In a similar fashion, no conclusion can be drawn on the effects of 2i or serum media on the accumulation of SNVs or SVs (lines 147-149), because this requires 2 consecutive clonal steps and a period in between to allow cells to accumulate mutations. This sentence and other comments on private SNVs/SVs can better be removed from the manuscript.

We have addressed this above, and respectfully disagree that our findings relating to private SNVs and SVs should be removed from the manuscript. The experiment requested by Reviewer #1, expansion of single-cell miPSC clones in 2i, is not currently possible. This type of analysis could possibly, although not certainly, be *best* done with 2 clonal expansion steps but that does not mean it can *only* be done in this way. This is especially the case as, again, clonal expansion from single miPSCs in 2i conditions is not possible in practice, even if it may be feasible for the adult stem cells referenced by Reviewer #1 above.

The RT inhibitor lamivudine does not inhibit programming. The study would be more interesting if shown that lamivudine leads to fewer retrotransposition events.

We worked extensively to experimentally address this point on revision. The original submission reported fewer retrotransposition events in miPSCs treated with lamivudine (3TC) during their reprogramming than those not treated with 3TC, where endogenous retrotransposition was detected via ONT sequencing. However, we were reluctant to make any strong conclusions from these data, given the experiment was n=2 (two 3TC-treated miPSC lines, two control miPSC lines). In the revision, we have added an additional 8 miPSC lines to make this experiment n=6 (6 3TC-treated, 6 controls) and thereby revealed a significant reduction in retrotransposition events. As we now state in the revised Results (line 263): “An additional 43 TE insertions were each detected in only one miPSC line and not the parental MEFs, or in the earlier Illumina sequencing, and were supported by at least one spanning ONT read (**Fig. 3d** and **Supplementary Table 3**). Performing PCR validation of these insertions, a step mainly intended to exclude them being polymorphisms carried by the parental or feeder MEF populations, we could amplify one (miPSC_50_B1) in the parental MEFs (**Extended Data Fig. 8b**). The remaining 42 putative *de novo* events comprised 16, 4 and 2 L1 T_F, G_F, and A insertions, respectively, as well as 5 and 13 SINE B1 and B2 insertions, respectively, and 2 ERV insertions. One L1 T_F insertion (miPSC_69_L1) PCR amplified in multiple miPSC lines (**Extended Data Fig. 8c**) and one L1 T_F insertion (miPSC_87_L1) amplified only in the miPSC line where it was detected by ONT sequencing (**Fig. 3e**). While the remaining insertions could not be PCR amplified in any sample, all 40 of the putative *de novo* L1 and SINE insertions carried clear TPRT hallmarks (**Fig. 3f** and **Supplementary Table 3**) and on this basis we considered them *bona fide* retrotransposition events. As well, both ERVs corresponded to the mobile intracisternal A-particle (IAP) family, presented a typical proviral structure of two long terminal repeats flanking an internal coding sequence, and generated TSDs of the expected size (6bp)^{80,81} (**Supplementary Table 3**). It was not clear whether the difference in (two) *de novo* ERV insertions being found here by ONT sequencing, and none by the earlier Illumina sequencing of different miPSC lines,

was due to chance or unknown technical reasons. Significantly fewer ($p < 0.02$, two-tailed t test) putative reprogramming-associated L1-mediated insertions were found on average in the 3TC-treated miPSCs (~1.3 per line) than in the control miPSCs (~5.2 per line) (**Fig. 3g**), consistent with L1 inhibition by 3TC (**Fig. 3b**). Overall, detection of endogenous retrotransposition events in bulk miPSCs by ONT sequencing yielded results orthogonal and complementary to our short-read genomic analyses.”

The authors also performed Nanopore sequencing of 4 bulk iPS cell lines and detected 16 TE insertions, one of which was also present in the feeders and one in the parental MEFs, leaving 14 de novo events. If I understand correctly, none of these events could be validated by PCR (0/14), while the success-rate for the de novo events that had been detected with other methods (mRC-seq + WGS) was much higher (38/41). Please explain this discrepancy and use a second method (PCR or otherwise) to validate the events detected by ONT.

This experiment was based on two principles. Firstly, as we state on line 242 of the Results: “A single DNA sequencing read, if of sufficient length and quality, can completely resolve a *de novo* TE insertion present in a heterogeneous cell population, as well its genomic flanks and accompanying TPRT hallmarks⁷⁷.” The point being here that PCR-free ONT sequencing can on its own discriminate genuine TE insertions, as these usually carry sequence hallmarks, such as target site duplications (TSD) associated with the TPRT (target-primed reverse transcription) retrotransposon integration mechanism. We established this ‘one spanning ONT read is enough’ principle in our previous work challenging the genomic integration of SARS-CoV-2 viral sequences (PMID: 34380018). To highlight this approach further, we applied ONT sequencing to HeLa cells carrying the mouse L1-mCherry retrotransposition reporter, as we now mention on line 246: “As a proof-of-principle before analyzing miPSCs, we applied PCR-free ONT sequencing to 5 HeLa cell lines expanded from single colonies (~5× genome-wide depth per colony) harboring mouse L1-mCherry retrotransposition events (**Supplementary Table 1**). Using the TLDR long-read TE insertion detection pipeline³⁵, we identified 41 L1-mCherry insertions spanned by at least one ONT read (**Supplementary Table 3**). L1-mCherry insertions overwhelmingly bore TPRT hallmarks regardless of whether they were detected by one, or more than one, ONT read (**Fig. 3c**), showing that single spanning ONT reads can reliably recover *bona fide* retrotransposition events.”

Secondly, PCR amplification of TE insertions is in this case primarily intended to exclude those events being mis-annotated genetic polymorphisms carried by the parental MEFs, or by feeder MEFs. Whilst in this experiment successful PCR amplification of a *de novo* TE insertion in the miPSC line would suggest they are relatively prevalent in that population, it is not required to show the event is real, which is already demonstrated by the ONT sequencing. To make this second part more clear, we have amended the Results (line 265) to state: “Performing PCR validation of these insertions, a step mainly intended to exclude them being polymorphisms carried by the parental or feeder MEF populations, we could amplify one (miPSC_50_B1) in the parental MEFs (**Extended Data Fig. 8b**).” By greatly expanding the number of bulk miPSC lines analyzed by ONT sequencing, from 4 to 12, the probability of a TE insertion being carried by the parental or feeder MEFs and only found in one of the

miPSC lines was reduced. Finally, in the expanded list of *de novo* insertions, we were able to PCR amplify 2 events in the miPSC lines they were detected in by ONT sequencing, which we note on line 270 of the Results: “One L1 T_F insertion (miPSC_69_L1) PCR amplified in multiple miPSC lines (**Extended Data Fig. 8c**) and one L1 T_F insertion (miPSC_87_L1) amplified only in the miPSC line where it was detected by ONT sequencing (**Fig. 3e**).” Although we didn’t consider this level of support to be essential, it nonetheless reassures readers that the approach is able to recover genuine retrotransposition events.

In line with this, were flanking genomic sequences detected by ONT? In other words, are these real transposition events or could it be reverse transcribed RNA that has not integrated?

The *de novo* TE insertions called by TLDR using PCR-free ONT sequencing carried sequence hallmarks in their flanking regions consistent with retrotransposon integration into the genome via TPRT. As noted in the above response, this is the main reason to consider these events to not be the result of another process, for example ectopic reverse transcription. We have added a new figure panel (**Fig. 3f**) and note this on line 272 of the Results: “While the remaining insertions could not be PCR amplified in any sample, all 40 of the putative *de novo* L1 and SINE insertions carried clear TPRT hallmarks (**Fig. 3f** and **Supplementary Table 3**) and on this basis we considered them *bona fide* retrotransposition events. As well, both ERVs corresponded to the mobile intracisternal A-particle (IAP) family, presented a typical proviral structure of two long terminal repeats flanking an internal coding sequence, and generated TSDs of the expected size (6bp)^{80,81} (**Supplementary Table 3**).”

The authors mention several differences between mice and men. With those differences in mind, how do the observations translate to human cells?

This is an important point and one of the main motivations for the present study, as we had previously shown that hiPSCs support endogenous retrotransposition (PMID: 26743714) whilst a previous work had claimed that, by contrast, miPSCs do not support this phenomenon (PMID: 21982236). In the original submission, we stated in the Introduction (line 52) that: “Engineered and endogenous L1 retrotransposition are supported by hiPSCs and ESCs^{45,50–52}.” and in the Discussion (line 339) that “Previous experiments employing hiPSCs and mouse and human ESCs showed L1 de-repression and mobilization were likely to take place in pluripotent cells^{34,41,42,45,46,48,50–52,89}.” To reinforce this message of concordance and translation to human cells, we now conclude the Introduction (line 59) by stating “Here, we analyze a diverse panel of miPSC genomes with short- or long-read sequencing and, as reported for hiPSCs and other pluripotent human cells^{45,50–52}, we detect numerous *de novo* TE insertions acquired by miPSCs.”

Minor comments:

In the abstract it is said that mouse iPSCs are able to silence transposable elements. This statement seems to be in conflict with some other comments in the manuscript: e.g. in the final paragraph of the introduction: “Mouse ESCs cultured in standard media containing serum express endogenous L1 proteins and support engineered L1 mobilisation. Naïve ESCs grown in media containing two small-molecule kinase

inhibitors (2i) in place of serum also exhibit L1 promoter hypomethylation.” Possibly better to rephrase the abstract.

The abstract is worded appropriately, as it reflects the available published literature, i.e. the work of Quinlan et al. (PMID: 21982236), where miPSCs were *reported* to silence transposable elements. This earlier work’s strong conclusion for retrotransposon silencing in miPSCs is in contrast to data from mouse ESCs, as Reviewer #1 notes here, and obviously is at odds with our findings.

I do not see the added value of the bulk experiments over the clonal analyses and would remove those from the manuscript.

We respectfully disagree, and have addressed this point regarding clonal expansion above.

In figure 1g only iPS cell lines are shown. This figure would improve if data of the parental cells is also included (the somatic cells after isolation prior to reprogramming) as a control for the bisulfite conversion.

Fig. 1g shows bisulfite sequencing analyses of the 5’UTR of 3 *de novo* L1 insertions, each present in one miPSC line. By definition (*de novo*), these L1 insertions are absent from the parental cells and for this reason the experiment requested here by Reviewer #1 does not make sense and cannot be performed. To address the gist of the point however, we now make reference to previous work (PMID: 29643204) measuring L1 methylation in somatic cells on line 219 of the Results: “By contrast, in MEFs, 83.6% of CpG dinucleotides in L1 T_F promoter monomers genome-wide were methylated, compared to 45.2% among the A67 and A172 miPSC lines (**Extended Data Fig. 6**), and resembled L1 T_F promoter methylation reported elsewhere for differentiated primary cells^{48,61}.”

In figure 1h a retrotransposition assay is conducted. This data is rather confusing, because 2 different cell types are presented in the same graph (HeLa + iPS cells) that cannot be compared with one another (see also discussion on species differences). In these experiments, cells are transfected with a reporter for retrotransposition events. According to the legend, the data has been normalised to L1 (in HeLa cells). Shouldn’t the data of the iPS cells be presented separately and normalised against L1 in iPS cells instead of the L1 in HeLa cells? Moreover, in the methods section, the experiments are being discussed for the HeLa cells, but not the iPSCs.

The retrotransposition assays shown in **Fig. 1h** were only conducted in HeLa cells, as noted in the figure legend. This is also why the experiments are not discussed for iPSCs in the Methods; the experiments were not performed in iPSCs, only HeLa cells. The goal of these experiments was to show that the *de novo* L1 insertions in miPSCs were generated by source L1s with functional ORFs. To make this clearer, the corresponding sentence of the revised Results (line 130) now reads: “As evidence of their *in vitro* mobility, miPSC_1_L1 and miPSC_4_L1 retrotransposed efficiently in HeLa cells, when expressed from their native promoter alone or with the addition of a cytomegalovirus promoter (**Fig. 1h**).”

The rationale between the two different types of media (2i versus serum) is not entirely clear

We agree this needed to be explained further and appreciate the suggestion. 2i and serum are the two main culture conditions used in the miPSC field, with the former considered to produce a more naïve stem cell than the latter. As stated on line 50 of the Introduction: “Naïve ESCs grown in media containing two small-molecule kinase inhibitors (2i) in place of serum also exhibit L1 promoter hypomethylation^{37,48,49}.” To set the rationale for testing both conditions further, we now state on line 141: “To assess the impact of standard and naïve culture conditions, respectively, upon L1 activity, each clone was divided and then further expanded in serum or 2i until p6 (**Fig. 2a**).”

Lines 152-154: Why were some events detected by mRC-seq and not by WGS and vice versa? Were the mRC-seq only events visible in the WGS data e.g. in IGV?

As stated on line 143 of the Results: “We then applied ~41× average genome-wide depth Illumina WGS and mRC-seq to miPSC single-cell clones 1-9, and mRC-seq only to clones 10-18, with each clone analyzed after culture in serum or 2i media (**Fig. 2a, Extended Data Fig. 1 and Supplementary Table 1**).” The 6 events detected by mRC-seq only were in one of the second set (numbered 10-18) of clones, while the 2 found by WGS were of an unusual type that is harder to detect with mRC-seq. We now explain this further on line 159 of the Results: “Note that the 6 events found only by mRC-seq were detected in the 9 miPSC clones (numbered 10-18) not analyzed with WGS. The 2 insertions found by WGS alone were both moderately 5' truncated and carried a 3' transduction, two features that reduced their probability of detection by mRC-seq, where enrichment probes target the 5' and 3' ends of L1 consensus sequences^{28,61}.”

Reviewer #2 (Remarks to the Author):

In this manuscript, Gerdes et al., study the role of retrotransposons as main drivers of the mutational landscape in mouse-induced pluripotent stem cells (miPSCs). This is a very interesting and comprehensive study that combines state-of-the-art methods in order to re-evaluate the role of retrotransposon integration in miPSCs. It is of importance to the field of transposable elements and regenerative medicine. The scientists reprogrammed mice that expressed a doxycycline-inducible reprogramming cassette. They used tissues of all 3 germ layers in order to produce miPSCs and looked for TE insertions by Illumina sequencing using WGS and mRC-seq. They also characterised the de novo inserted TE sequences. Additionally, the researchers used single-cell isolated clones for cell expansion to look at the TE de novo insertions in a more homogenous cell population. They found a polymorphic retrotransposition competent L1 that evades DNA methylation. Interestingly, inhibiting the L1 reverse transcription with lamivudine to prevent potential de novo mutations had no effect on reprogramming activity. Finally, Gerdes et al confirmed genomic TE insertions with nanopore sequencing in bulk miPSCs. Nanopore sequencing also allowed them to look at genome-wide DNA demethylation during iPSC reprogramming and the researchers found young L1s to lose more methylation than other sites.

We thank Reviewer #2 for their positive and thoughtful remarks.

Major comments:

- I am not sure how the scientists evaluated the reprogramming efficiency of the miPSCs that were treated with lamivudine or not. The scientists should also evaluate the general demethylation dynamics in the lamivudine treated samples (similar to Fig 3c) in order to see whether epigenetic reprogramming is at all affected.

There are two points here, both of which we agreed were important to address. Firstly, we evaluated more extensively whether lamivudine (3TC) affects reprogramming efficiency. We have now greatly expanded **Extended Data Fig. 7** to incorporate data showing that the prevalence of pluripotency markers Oct4-GFP, Nanog and alkaline phosphatase (AP) is no different in miPSCs reprogrammed with or without 3TC; reprogramming efficiency is unaffected by 100 μ M 3TC. We note these data on line 233 of the Results: “By performing titration experiments to optimize the use of 3TC during miPSC generation, we determined that 3TC concentrations of up to 100 μ M did not reduce viability of cultured MEFs or miPSCs (**Extended Data Fig. 7a**) or MEF reprogramming efficiency (**Fig. 3a** and **Extended Data Fig. 7b-e**).” Secondly, we analyzed the methylation landscape of miPSCs cultured with and without 3TC, now expanded to n=6 of each condition, and found no impact of lamivudine upon global or retrotransposon methylation. These results are shown in the revised **Fig. 4**. We mention this analysis on line 301 of the Results: “By contrast, we observed no significant local or global differences in L1 T_F or protein-coding gene promoter methylation between control and 3TC-treated miPSCs (**Fig. 4** and **Supplementary Table 5**).”

- In the nanopore sample Gerdes et al. still found 2 L1s that are able to evade RT inhibition with lamivudine. This needs to be added to the discussion and also rephrased

in the abstract, because of course if inhibition of L1 RT is not sufficient the question is whether reprogramming is affected if you were to inhibit L1 RT to 100%. The researchers state that this result is not statistically significant because of the low sample number however the question is also the depth of the nanopore sequencing. Does this allow to catch all the L1s that are inserted de novo?

These are good points as well, thank you. Our existing L1 reporter data (**Fig. 3b**) suggested 100 μ M lamivudine (3TC) would nearly abolish retrotransposition in cultured cells. On revision, we greatly expanded the number of miPSC lines analyzed with ONT sequencing (n=6 3TC, n=6 control) and thus we determined 3TC also significantly reduced endogenous retrotransposition during reprogramming (**Fig. 3g**). We note the expanded miPSC cohort on on line 254 of the Results: “Next, to survey endogenous retrotransposition in miPSCs, we ONT sequenced (~20 \times average genome-wide depth) 6 bulk miPSC lines reprogrammed in the presence of 100 μ M 3TC, 6 control miPSC lines not treated with 3TC, and matched parental MEFs (**Fig. 3a** and **Supplementary Table 1**).” We describe the analysis of these additional miPSC lines on line 263 of the Results: “An additional 43 TE insertions were each detected in only one miPSC line and not the parental MEFs, or in the earlier Illumina sequencing, and were supported by at least one spanning ONT read (**Fig. 3d** and **Supplementary Table 3**). Performing PCR validation of these insertions, a step mainly intended to exclude them being polymorphisms carried by the parental or feeder MEF populations, we could amplify one (miPSC_50_B1) in the parental MEFs (**Extended Data Fig. 8b**). The remaining 42 putative *de novo* events comprised 16, 4 and 2 L1 T_F, G_F, and A insertions, respectively, as well as 5 and 13 SINE B1 and B2 insertions, respectively, and 2 ERV insertions. One L1 T_F insertion (miPSC_69_L1) PCR amplified in multiple miPSC lines (**Extended Data Fig. 8c**) and one L1 T_F insertion (miPSC_87_L1) amplified only in the miPSC line where it was detected by ONT sequencing (**Fig. 3e**). While the remaining insertions could not be PCR amplified in any sample, all 40 of the putative *de novo* L1 and SINE insertions carried clear TPRT hallmarks (**Fig. 3f** and **Supplementary Table 3**) and on this basis we considered them *bona fide* retrotransposition events. As well, both ERVs corresponded to the mobile intracisternal A-particle (IAP) family, presented a typical proviral structure of two long terminal repeats flanking an internal coding sequence, and generated TSDs of the expected size (6bp)^{80,81} (**Supplementary Table 3**). It was not clear whether the difference in (two) *de novo* ERV insertions being found here by ONT sequencing, and none by the earlier Illumina sequencing of different miPSC lines, was due to chance or unknown technical reasons. Significantly fewer (p<0.02, two-tailed t test) putative reprogramming-associated L1-mediated insertions were found on average in the 3TC-treated miPSCs (~1.3 per line) than in the control miPSCs (~5.2 per line) (**Fig. 3g**), consistent with L1 inhibition by 3TC (**Fig. 3b**). Overall, detection of endogenous retrotransposition events in bulk miPSCs by ONT sequencing yielded results orthogonal and complementary to our short-read genomic analyses.” Please note we also discuss ONT sequencing depth in one of the below answers. We think these reinforced results are in line with the abstract, but we also feel Reviewer #2’s comment about still finding *de novo* TE insertions in the miPSC lines treated with 3TC was important. Our view is that these events likely predate reprogramming and we have redrafted the concluding part of the Discussion (line 364) to bring this to the fore: “Fortunately, strategies to minimize TE-mediated mutagenesis, including via the use of 3TC or another L1

reverse transcriptase inhibitor⁷⁵, appear achievable without affecting DNA methylation or reprogramming efficiency. While the frequency of *de novo* endogenous L1-mediated insertions found in miPSCs was significantly reduced by 3TC, a handful of events, likely arising prior to reprogramming, were nonetheless observed in 3TC-treated miPSCs. Reverse transcriptase inhibitors thus may be incorporated into future miPSC derivation protocols to attenuate *de novo* retrotransposition, in addition to genomic screening of miPSC lines.”

- I would like the scientists to comment on the fact that they did not find ERV elements in the short-read data but found them in the long read data.

We looked into this at length and the short answer is we don't have a good explanation for this result. With the additional 8 ONT sequenced miPSC lines, one of the three original ERVs was re-annotated as not being *de novo* during reprogramming as it was now found in two miPSC lines in total. The remaining two *de novo* ERV insertions, both IAPs, carry all the hallmarks of LTR retrotransposition and were still found in just one miPSC line each. As there were only two of these events, we cannot reasonably explain this result as not simply being an outcome of chance. We comment briefly to this effect on line 275 of the Results: “As well, both ERVs corresponded to the mobile intracisternal A-particle (IAP) family, presented a typical proviral structure of two long terminal repeats flanking an internal coding sequence, and generated TSDs of the expected size (6bp)^{80,81} (Supplementary Table 3). It was not clear whether the difference in (two) *de novo* ERV insertions being found here by ONT sequencing, and none by the earlier Illumina sequencing of different miPSC lines, was due to chance or unknown technical reasons.”

- What is the depth of the nanopore sequencing and what is the exact overlap between nanopore sequencing and WGS or RC-Seq? Did you only take forward reads that overlap both WGS and had at least 1 read in nanopore? Do you think there are more insertions that you missed because of the depth of nanopore sequencing which could not be found in WGS and RC-seq?

Illumina WGS (and mRC-seq) and nanopore (ONT) WGS were applied to mutually exclusive miPSC line cohorts, with ONT sequencing used to analyze 12 bulk miPSC lines derived from MEFs (n=6 lamivudine treated, n=6 control) and Illumina sequencing applied to an earlier cohort of 18 single-cell miPSC clones reprogrammed from MEFs. There was therefore no overlap in terms of *de novo* insertions found by both Illumina and ONT sequencing (and actually these would not be annotated as *de novo* if they had been found by both approaches, in multiple miPSC lines). The major advantage of ONT sequencing over Illumina sequencing is that a single spanning ONT read is sufficient to completely resolve insertions, whereas Illumina reads rarely span insertions, meaning the ONT-based approach is much more sensitive for *de novo* retrotransposition events than Illumina sequencing. We now lay this out in the Results, line 242: “A single DNA sequencing read, if of sufficient length and quality, can completely resolve a *de novo* TE insertion present in a heterogeneous cell population, as well its genomic flanks and accompanying TPRT hallmarks⁷⁷. Long-read sequencing, as for example developed by Oxford Nanopore Technologies (ONT), is well suited to this application, and can locate TE insertions within repetitive genomic regions refractory to short-read methods^{35,77-79}. As a proof-of-principle before analyzing miPSCs, we applied

PCR-free ONT sequencing to 5 HeLa cell lines expanded from single colonies (~5× genome-wide depth per colony) harboring mouse L1-mCherry retrotransposition events (**Supplementary Table 1**). Using the TLDR long-read TE insertion detection pipeline³⁵, we identified 41 L1-mCherry insertions spanned by at least one ONT read (**Supplementary Table 3**). L1-mCherry insertions overwhelmingly bore TPRT hallmarks regardless of whether they were detected by one, or more than one, ONT read (**Fig. 3c**), showing that single spanning ONT reads can reliably recover *bona fide* retrotransposition events.” To address Reviewer #2’s questions about ONT sequencing depth and sensitivity in a different way, we asked what fraction of the non-reference TE insertions found in the parental MEFs were found in each miPSC line. On line 254 of the Results we now state: “Next, to survey endogenous retrotransposition in miPSCs, we ONT sequenced (~20× average genome-wide depth) 6 bulk miPSC lines reprogrammed in the presence of 100μM 3TC, 6 control miPSC lines not treated with 3TC, and matched parental MEFs (**Fig. 3a** and **Supplementary Table 1**). Using these data, TLDR identified 3,975 non-reference TE insertions carried by the parental MEFs (**Supplementary Table 4**). Of these, 3,429 (86.3%) corresponded to previously known insertions⁵⁶ and 99.6% were found in each miPSC line, on average (**Supplementary Table 4**).” It is of course likely that additional ONT sequencing would have revealed more *de novo* insertions; the key point here is that lamivudine appeared to greatly reduce the frequency of these events at the depth achieved here.

- Does the reprogramming efficiency differ for the different cell types? The scientist should report the numbers of original reprogramming efficiency from their FACS sort of GFP positive cells. Does this lead to a differing number of L1s?

We thank Reviewer #2 for this excellent suggestion. We had not considered reprogramming efficiency as a potential arbiter of *de novo* retrotransposition rate. As requested, we have amended **Fig. 1a** to now display the reprogramming efficiencies (*Oct4*-GFP⁺ %) for each primary cell type and the number of *de novo* L1 insertions found in each. Indeed, more insertions (6/7) were found in cells generating the lower half of reprogramming efficiencies than those in the top half (1/7). With small counts this difference was however not statistically significant (p<0.13, binomial test). We note the outcome in the Results (line 113) as we found this an interesting observation: “In sum, 10/26 miPSC lines harbored at least one PCR validated *de novo* L1 insertion. Not counting the mosaic miPSC_2_L1 insertion, miPSCs from all 3 animals and 4/9 primary cell types, representing each germ layer, presented at least one *de novo* L1 insertion (**Fig. 1a** and **Supplementary Table 3**). Of these insertions, 4/7 were detected in astrocyte-derived miPSCs and 6/7 in miPSCs obtained from primary cells in the bottom 50% of reprogramming efficiencies (**Fig. 1a**), though neither of these proportions were statistically significant (binomial test).”

Reviewer #3 (Remarks to the Author):

General comments:

The authors focus on a possible drawback of the use of somatic cell reprogramming to derive mouse-induced PSCs which was never fully addressed before: the vulnerability of miPSCs to TE-mediated mutagenesis due to TE mobility during reprogramming. Mutagenesis of miPSCs could impair their applications in many fields thus, it is important to understand and investigate all the possible sources of reprogramming-derived/mediated mutagenesis and the possibility to counteract. I liked that the authors addressed these points in their work thus I value positively this paper, but more effort in addressing the Lamivudine data would give more strength, in my opinion. In general, I find the paper to be clear and linear: the results follow a logical flow, I think they are well described in the text and by the figures, captions are very well written (all details are present). Methods are exhaustive. Discussion highlights the future steps that can be taken.

We thank Reviewer #3 for their supportive review and feedback.

However, I can't really comment too much on the novelty and the strength of the techniques used, I would add in the introduction a conclusive section that describes what the authors found and their contribution to the field.

We have added a concluding sentence, as requested, to the Introduction (line 59) by stating: "Here, we analyze a diverse panel of miPSC genomes with short- or long-read sequencing and, as reported for hiPSCs and other pluripotent human cells^{45,50-52}, we detect numerous *de novo* TE insertions acquired by miPSCs." We agreed this was good to emphasize but made the additional text concise, adhering to editorial guidelines asking to avoid redundancy between the Introduction and later sections.

Other comments:

1. In figure 1A I would change the name of the mouse model to make it clearer.

We agreed and changed the name to "Doxycycline-inducible reprogrammable (OKSM) mouse model" to clearly define what the mouse model is in Fig. 1.

2. PCR controls in figure 1: would it be good to add as negative control the parental line and not MEFs only?

This was a good question and something we needed to explain better. A technical caveat of the approach is that we wanted complete sets of 9 miPSC lines, one from each primary cell type, from each animal, despite reprogramming being variable and in some cases quite low in efficiency. This required a tough judgment call. We opted to not reserve an aliquot of primary cells for DNA extraction, trying to obtain as many miPSCs as possible to satisfy the minimum genomic DNA input requirements for downstream sequencing. Even so, for one of the 3 animals we ultimately did not have enough genomic DNA for Illumina sequencing (why one of the three animals, A172, does not have an astrocyte-derived miPSC line). In this case, the ability to call a retrotransposition event as being private to one miPSC line rests on it not being found by PCR in the other miPSC lines. We felt this was a reasonable

assumption, and was necessary to complete the bulk miPSC genomic analysis. To explain this further to readers, we have added to the legend for **Fig. 1** the following text: “**Note: given the variable, and in some cases very low, reprogramming efficiencies shown in panel (a), and the objective to obtain a full set of 9 miPSC lines from each animal, we entirely used each sorted primary cell population for reprogramming, relying on PCR amplification in a single miPSC line to validate reprogramming-associated retrotransposition events.**”

3. Figure 1C: there are some faint bands. Are they no-specific bands or what?

The very faint smear in the other lanes is a non-specific product from the empty/filled validation PCR reaction. This insertion was located within an older L1 insertion and therefore the PCR tended to generate off-target bands, as well as the on-target product. The off-target product was orders of magnitude less bright than the on-target product and somewhat smaller. We have amended the legend for **Fig. 1c** to include the following statement: “**Note: the very faint, smaller-sized gel band observed in most of the samples was an off-target product.**” Also below we show a gel image obtained from an additional PCR amplification experiment for this insertion showing the on-target product (red arrow) in the miPSC line (A67_7 astrocyte) where the insertion was detected by the genomic analysis. The off-target product is absent.

4. They say Lamivudine doesn't affect the reprogramming efficiency, however, to support this claim they provide only the data about cell survival (supplementary figure 7). After the reprogramming, I think the quantification of OCT4+ cells and additional markers would be preferable to support such a conclusion.

We agreed fully with this suggestion. We have greatly expanded **Extended Data Fig. 7** to incorporate data showing that the prevalence of pluripotency markers Oct4-GFP, Nanog and alkaline phosphatase (AP) is no different in miPSCs reprogrammed with or without

lamivudine (3TC), and reprogramming efficiency is not affected at 100 μ M 3TC. We note these data on line 233 of the Results: “By performing titration experiments to optimize the use of 3TC during miPSC generation, we determined that 3TC concentrations of up to 100 μ M did not reduce viability of cultured MEFs or miPSCs (**Extended Data Fig. 7a**) or MEF reprogramming efficiency (**Fig. 3a** and **Extended Data Fig. 7b-e**).”

5. They didn't observe any significantly in the difference between TE transpositions after reprogramming with or without Lamivudine, however, they state that there might be an effect of Lamivudine. I'm suggesting making this result more redundant and stronger. Either repeat the experiment (to increase the N) or try different strategies. It would be a very cool result and strengthen the entire work.

We focused our efforts on addressing this point. We added 4 miPSC lines to each of the control and 3TC-treated datasets, making n=6 in each group, as noted on line 254 of the Results: “Next, to survey endogenous retrotransposition in miPSCs, we ONT sequenced (~20 \times average genome-wide depth) 6 bulk miPSC lines reprogrammed in the presence of 100 μ M 3TC, 6 control miPSC lines not treated with 3TC, and matched parental MEFs (**Fig. 3a** and **Supplementary Table 1**).” Analyzing this expanded cohort, the reduction in L1-mediated retrotransposition in the 3TC-treated miPSCs was now significant ($p < 0.02$). We added to **Fig. 3** to build upon this point and the analysis further on line 263 of the Results: “An additional 43 TE insertions were each detected in only one miPSC line and not the parental MEFs, or in the earlier Illumina sequencing, and were supported by at least one spanning ONT read (**Fig. 3d** and **Supplementary Table 3**). Performing PCR validation of these insertions, a step mainly intended to exclude them being polymorphisms carried by the parental or feeder MEF populations, we could amplify one (miPSC_50_B1) in the parental MEFs (**Extended Data Fig. 8b**). The remaining 42 putative *de novo* events comprised 16, 4 and 2 L1 T_F, G_F, and A insertions, respectively, as well as 5 and 13 SINE B1 and B2 insertions, respectively, and 2 ERV insertions. One L1 T_F insertion (miPSC_69_L1) PCR amplified in multiple miPSC lines (**Extended Data Fig. 8c**) and one L1 T_F insertion (miPSC_87_L1) amplified only in the miPSC line where it was detected by ONT sequencing (**Fig. 3e**). While the remaining insertions could not be PCR amplified in any sample, all 40 of the putative *de novo* L1 and SINE insertions carried clear TPRT hallmarks (**Fig. 3f** and **Supplementary Table 3**) and on this basis we considered them *bona fide* retrotransposition events. As well, both ERVs corresponded to the mobile intracisternal A-particle (IAP) family, presented a typical proviral structure of two long terminal repeats flanking an internal coding sequence, and generated TSDs of the expected size (6bp)^{80,81} (**Supplementary Table 3**). It was not clear whether the difference in (two) *de novo* ERV insertions being found here by ONT sequencing, and none by the earlier Illumina sequencing of different miPSC lines, was due to chance or unknown technical reasons. Significantly fewer ($p < 0.02$, two-tailed t test) putative reprogramming-associated L1-mediated insertions were found on average in the 3TC-treated miPSCs (~1.3 per line) than in the control miPSCs (~5.2 per line) (**Fig. 3g**), consistent with L1 inhibition by 3TC (**Fig. 3b**). Overall, detection of endogenous retrotransposition events in bulk miPSCs by ONT sequencing yielded results orthogonal and complementary to our short-read genomic analyses.”

6. Would it be possible to understand the probability of L1 insertions in gene exons or gene regulatory elements based on the frequency of Degenerate L1 endonucleases motifs within the genome? Thus, to comment on the real probability that L1 transpositions can really generate functional mutations?

This was a good point to consider further. Our thinking here is that L1 insertions into exons (and regulatory elements like enhancers) are likely to alter gene function and expression but also represent a very small fraction of events. Intronic L1 insertions, by comparison, are far more common and there is good evidence to suggest these can substantially perturb gene expression (e.g. PMID: 15152245), even if they do not tend to have the dramatic effects of exonic mutations. In the Discussion (line 354), we now state: “Exonic retrotransposon insertions are clear in their potential to cause disease⁹³. However, the L1 endonuclease does not favor exons^{65,66}, which are depleted for its AT-rich recognition motif and make up only ~2% of the genome⁹⁴, and none of the 81 *de novo* L1-mediated insertions reported here in miPSCs were exonic. By comparison, L1 integration within introns occurs much more frequently. We observed 30/81 (37.0%) L1-mediated insertions in introns, in line with the proportion of the genome occupied by these regions^{65,66}. Whilst intronic events are less likely to be pathogenic than exonic insertions, they can perturb gene expression, for example through reduced transcriptional elongation⁹⁵ or, as shown here, the provision of new promoter elements⁹⁶. Retrotransposition into the introns of protein-coding genes, as observed here for *Brca1* and *Dmd*, could therefore undermine miPSC models of human disease. Such mutations necessitate screening of miPSC lines⁴.”

REVIEWER COMMENTS

Reviewer #1 (Remarks to the Author):

The revisions by Gerdes et al. have greatly improved the manuscript. My main concern with the previous version was that I was not convinced that the observed events were truly de novo, because these could also be low frequent events in the parental population of somatic cells that were used for reprogramming. Nevertheless, my concern has been largely taken away by the ONT experiments where the authors show that 3TC treatment during reprogramming reduces the number of TE events. I am still of the opinion that a clonal step prior to reprogramming would be highly informative (for example using the bulge or intestinal stem cells depicted in Fig. 1a), because this allows discriminating between rare somatic events (SNVs, SVs, and TE insertions) and events as a result of the reprogramming process. However, this is for me no longer a major issue that should impede further processing of this manuscript.

Comments:

I see limited added value for the 2i experiments. My advice would be to take this comparison out of the manuscript. As I mentioned in my previous comments, it would have been more informative if a second clonal step had been performed to see if there is higher TE activity in one condition than the other. The authors claim that clonal steps in 2i are not possible, which I sincerely doubt and also overcome by first switching back to serum conditions.

The ONT experiments where the authors show that 3TC treatment during reprogramming reduces the number of TE events deserve more attention, for example in the abstract.

3TC is a nucleoside analog, which may have mutagenic potential

<https://doi.org/10.1016/j.stem.2021.07.012>. The authors may wish to discuss this and examine their ONT data to see if 3TC treated iPSCs have similar or more mutations than untreated iPSCs.

The authors conclude that "Choice of primary cell type, at least among the diverse panel assembled here, may therefore not significantly impact the frequency of SNVs and SVs later found in miPSC lines." This is in contrast with a recent finding that shows that human skin derived iPSCs harbor more SBSs/DBSs as a result of UV exposure (<https://doi.org/10.1038/s41588-022-01147-3>). The authors may wish to discuss their results in the light of these recent findings so that ignorant readers do not get the impression that there are little differences between cell sources in preexisting mutational load.

Different methods were used to detect TE events. It is not entirely clear which of the 43 TE events in table 1 were captured with what methods (and thereby cross-validated). Would be nice if this were included in the table.

Related to that, in line 113 the authors mention that L1 insertions were found in 10/26 miPSC lines, but the numbers do not seem to add up (I apologize if I accidentally missed some): from the preceding text it seems only 8 were detected (4 with TE break, and 4 with mRC-seq), 6 of which were full length (lines 102-103), and one was very heavily 5' truncated (line 105) and the final was heavily 5' truncated and inverted (lines 106-107).

The abbreviation TSD (target site duplication) is only explained in the legend of fig1, not in the main text

Also provide the full text to TPRT (target-primed reverse transcription) at first use

Reviewer #2 (Remarks to the Author):

The authors addressed all my requests in full and added relevant experimental data to support their hypothesis.

Reviewer #3 (Remarks to the Author):

Thanks to the author to carefully revised their work. I have only one final comments.

To show that Lamivudine doesn't affect the reprogramming efficiency, they showed new data in figure Extended Data Fig. 7. These data show that the prevalence of pluripotency markers Oct4-GFP, Nanog and alkaline phosphatase (AP) is no different in miPSCs reprogrammed with or without lamivudine (3TC), and reprogramming efficiency is not affected at 100 μ M 3TC.

However, it is not clear why the authors did not show the histogram indicating the mean reprogramming efficiency (Oct4-GFP+) \pm SD as reported in Figure 1a for each primary cell type. This will provide more consistent quantification throughout their work. I can be wrong, but the efficiency showed in figure 7 is much larger than the one displayed for the cell lines. Is this causing an artefact on the evaluation of the effect on Lamivudine in high-efficient reprogramming conditions obtained for MEF?

Reviewer #1 (Remarks to the Author):

The revisions by Gerdes et al. have greatly improved the manuscript. My main concern with the previous version was that I was not convinced that the observed events were truly *de novo*, because these could also be low frequent events in the parental population of somatic cells that were used for reprogramming. Nevertheless, my concern has been largely taken away by the ONT experiments where the authors show that 3TC treatment during reprogramming reduces the number of TE events. I am still of the opinion that a clonal step prior to reprogramming would be highly informative (for example using the bulge or intestinal stem cells depicted in Fig. 1a), because this allows discriminating between rare somatic events (SNVs, SVs, and TE insertions) and events as a result of the reprogramming process. However, this is for me no longer a major issue that should impede further processing of this manuscript.

We thank Reviewer #1 for noting the improvements made to the manuscript, we appreciate this more positive standpoint very much.

Comments:

I see limited added value for the 2i experiments. My advice would be to take this comparison out of the manuscript. As I mentioned in my previous comments, it would have been more informative if a second clonal step had been performed to see if there is higher TE activity in one condition than the other. The authors claim that clonal steps in 2i are not possible, which I sincerely doubt and also overcome by first switching back to serum conditions.

Respectfully, we addressed this point in the first round of revision. We disagreed then and now that the 2i experiments should be removed but we did add text noting that additional variants *could* have been found if the approach proposed by Reviewer #1 was possible and adopted. As we said, in our hands, the miPSCs do not expand from single cells in 2i.

The ONT experiments where the authors show that 3TC treatment during reprogramming reduces the number of TE events deserve more attention, for example in the abstract.

We agreed that this was a good suggestion, although we had to alter the text somewhat to adhere to the abstract word limit. The penultimate Abstract sentence now reads: “Treatment with the L1 inhibitor lamivudine did not hinder reprogramming and efficiently blocked endogenous retrotransposition, as detected by ONT sequencing.”

3TC is a nucleoside analog, which may have mutagenic potential

<https://doi.org/10.1016/j.stem.2021.07.012>. The authors may wish to discuss this and examine their ONT data to see if 3TC treated iPSCs have similar or more mutations than untreated iPSCs.

The reference provided by Reviewer #1 (Hoeck, Tjoonk, van Boxtel and Cuppen, *BMC Cancer*, 2019) is for a study of ganciclovir, a different nucleoside analog to 3TC, and therefore it is not known whether 3TC has similar mutagenic potential or not. We will refrain from speculating on this point. Calling *de novo* TE insertions with our ONT data is feasible

because they carry TSDs and other TPRT hallmarks that can discriminate *bona fide* mutations and artifacts. *De novo* SNVs, by contrast, are very difficult to robustly call with ONT sequencing at this stage, and for this reason we have not performed this analysis.

The authors conclude that “Choice of primary cell type, at least among the diverse panel assembled here, may therefore not significantly impact the frequency of SNVs and SVs later found in miPSC lines.” This is in contrast with a recent finding that shows that human skin derived iPSCs harbor more SBSs/DBSs as a result of UV exposure (<https://doi.org/10.1038/s41588-022-01147-3>). The authors may wish to discuss their results in the light of these recent findings so that ignorant readers do not get the impression that there are little differences between cell sources in preexisting mutational load.

This sentence referred explicitly to the miPSC lines analyzed here, whereas the recent publication noted by Reviewer #1 deals with reprogrammed human cells, which are different for many reasons, including environmental exposure. We disagree that readers would get the wrong impression and our view is that the sentence, about the mouse cells used in our study, was correct as written and should remain as is.

Different methods were used to detect TE events. It is not entirely clear which of the 43 TE events in table 1 were captured with what methods (and thereby cross-validated). Would be nice if this were included in the table.

We saw the merit of this suggestion and have added the relevant information to Table 1.

Related to that, in line 113 the authors mention that L1 insertions were found in 10/26 miPSC lines, but the numbers do not seem to add up (I apologize if I accidentally missed some): from the preceding text it seems only 8 were detected (4 with TE break, and 4 with mRC-seq), 6 of which were full length (lines 102-103), and one was very heavily 5' truncated (line 105) and the final was heavily 5' truncated and inverted (lines 106-107). There were indeed 8 *de novo* L1 insertions. Seven of these were found in one miPSC line and were considered reprogramming-associated, and the other, miPSC_2_L1, was found in multiple miPSC lines derived from one animal and considered to precede reprogramming (a mosaic insertion). To make this accounting clearer, we have rephrased the relevant sentence (line 113) to read: “Including the mosaic miPSC_2_L1 insertion, 10/26 miPSC lines harbored at least one PCR validated *de novo* L1 insertion.”

The abbreviation TSD (target site duplication) is only explained in the legend of fig1, not in the main text

TSD/TSDs was already defined at first use in the revision, as “target site duplications” (now line 34).

Also provide the full text to TPRT (target-primed reverse transcription) at first use.

TPRT was also already defined at first use in the same sentence, as “target-primed reverse transcription” (now line 33).

Reviewer #2 (Remarks to the Author):

The authors addressed all my requests in full and added relevant experimental data to support their hypothesis.

Reviewer #3 (Remarks to the Author):

Thanks to the author to carefully revised their work. I have only one final comments.

To show that Lamivudine doesn't affect the reprogramming efficiency, they showed new data in figure Extended Data Fig. 7. These data show that the prevalence of pluripotency markers Oct4-GFP, Nanog and alkaline phosphatase (AP) is no different in miPSCs reprogrammed with or without lamivudine (3TC), and reprogramming efficiency is not affected at 100 μ M 3TC. However, it is not clear why the authors did not show the histogram indicating the mean reprogramming efficiency (Oct4-GFP+) \pm SD as reported in Figure 1a for each primary cell type. This will provide more consistent quantification throughout their work. I can be wrong, but the efficiency showed in figure 7 is much larger than the one displayed for the cell lines. Is this causing an artefact on the evaluation of the effect on Lamivudine in high-efficient reprogramming conditions obtained for MEF?

This was a good point, we thank Reviewer #3 for noticing this. We did not think to calculate and show the MEF reprogramming efficiency in Extended Data Fig. 7. We agreed with Reviewer #3 that doing so would make the analysis consistent with the other primary cell types shown in Figure 1a. We have now added these data as an additional histogram in Extended Data Fig. 7c. As before, there was no significant difference in reprogramming efficiency between control and 3TC conditions. In each case, the reprogramming efficiency was less than 2%. This was higher than most of the other cell types shown in Figure 1, but still relatively similar in absolute terms. MEFs reprogram well and are a workhorse primary cell type for miPSC generation, which is why we focused on MEF reprogramming to test the influence of 3TC and gain the most impactful insights for the field. Would 3TC work as well to inhibit L1 in miPSCs reprogrammed from other primary cell types? There is no reason to think it would not, given that it is an extremely efficient L1 inhibitor in other mammalian cells (e.g. PMID: 31155508). To be certain we would need to perform a much larger experiment involving the 9 primary cell types shown in Figure 1a, which we think would be fairly left to a future study.

REVIEWERS' COMMENTS

Reviewer #3 (Remarks to the Author):

I would like to thank the reviewers for their response. All my concerns are sorted.